# Decadal re-forecasts of glacier climatic mass balance

Larissa Nora van der Laan[1,2], Anouk Vlug[3,4], Adam A. Scaife[5,6], Fabien Maussion[4,7], and Kristian Förster[8,2]

[1]Niels Bohr Institute, University of Copenhagen, Copenhagen, Denmark
[2]Institute of Hydrology and Water Resources Management, Leibniz University Hannover, Hannover, Germany
[3]Institute of Geography, University of Bremen, Bremen, Germany
[4]Department of Atmospheric and Cryospheric Sciences, University of Innsbruck, Innsbruck, Austria
[5]Met Office Hadley Centre, Exeter, UK
[6]College of Engineering, Mathematics and Physical Sciences, University of Exeter, Exeter, UK
[7]Bristol Glaciology Centre, School of Geographical Sciences, University of Bristol, Bristol, UK
[8]Institute of Ecology and Landscape, University of Applied Sciences Weihenstephan-Triesdorf, Freising, Germany

**Correspondence:** Larissa Nora van der Laan (larissa.vdlaan@nbi.ku.dk)

**Abstract.**

We present the first study employing decadal re-forecasts to simulate global glacier climatic mass balance, bridging the gap between seasonal forecasts and long-term projections of glacier contributions to catchment hydrology and sea-level rise. Using the Open Global Glacier Model (OGGM) and Coupled Model Intercomparison Project Phase 6 (CMIP6) decadal re-forecasts of temperature and precipitation, we demonstrate the predictive skill of glacier mass balance re-forecasts over decadal time scales in two components: for a set of 279 reference glaciers, making use of their mass balance record, and all land-terminating glaciers, making use of the globally available geodetic mass balance, respectively. Results show that forcing OGGM with decadal re-forecasts outperforms persistence forecasts and historical General Circulation Model (GCM) simulations. Specifically, out of 279 reference glaciers, 174 show improved skill when forcing OGGM with decadal re-forecasts for decadal mean mass balance, and 186 show improved skill for cumulative mass balance. On a global scale, forcing with decadal re-forecasts yields the best agreement with observed regional mean mass balances for the period 2000–2020. These findings demonstrate moderate improvements from using decadal re-forecasts, though statistical significance is limited. While improvements are modest, the results suggest decadal re-forecasts may offer potential for improved near-term glacier predictions relevant for hydrological applications, particularly in regions where near-term forecasts can inform water resource management and climate adaptation strategies.

## 1 Introduction

As unique indicators of climate change, water storage reservoirs and culturally significant sites, glaciers serve a multitude of purposes (Allison, 2015; Bosson et al., 2019; Farinotti et al., 2020; Jansson et al., 2003). Observing and simulating their response to climate change on various time scales, from millennia to a focus on the past century, is an essential and continuously developing field of study (e.g. Goosse et al., 2018; Hock et al., 2019; Malles and Marzeion, 2021; Marzeion et al., 2017; Roe et al., 2021; Vargo et al., 2020). While the storage outside the Greenland and Antarctic Ice Sheets only constitutes a small

percentage of total global freshwater storage in ice, it is equivalent to approximately 0.32 m of sea level rise (Farinotti et al., 2019). Since these smaller ice bodies respond fastest to changes in the climate, glaciers (outside of the ice sheets) were the largest contributor to sea-level rise for most of the past century (Frederikse et al., 2020), overtaken by thermosteric contribution after 1970, and are expected to remain a significant contributor in the foreseeable future (Frederikse et al., 2020; Slangen et al., 2017).

Water is accumulated and released by glaciers on various time scales, ranging from long-term storage in ice and firn to short-term storage in snow cover. Within their basins, glaciers act as a buffering system, preventing precipitation from immediately turning into runoff in downstream rivers (Jansson et al., 2003). The seasonality of glacier runoff therefore modulates downstream flow, providing meltwater in otherwise potentially dry seasons or years of low flow (Huss and Hock, 2018; Ultee et al., 2022; Förster and van der Laan, 2022). Due to this buffering capacity, they are essential parts of global water towers, defined as mountain range water storage and supply to downstream communities and ecosystems, upon which 22 % of the global population is dependent for their water needs (Immerzeel et al., 2020).

With changes in climate, glacier mass balance – a temporal integration of both accumulation and melt, largely governed by temperature and precipitation – is altered, impacting over time the glacier mass and thus storage capacity. Despite their critical relevance, decadal time scales are rarely considered in glacier modeling studies. This omission is significant, given that such time scales are critical for water resource management, anticipating glacier change induced impacts on catchment hydrology (Frans et al., 2016; Lane and Nienow, 2019). The need for annual to decadal predictions is well recognized, despite the early stage of the field (Boer et al., 2016; Merryfield et al., 2020). When using the term decadal in this study, it encompasses time scales of one to ten years. The term "decadal prediction", as used here, encompasses predictions on annual, multi-annual and decadal time scales (Boer et al., 2016).

In 2016, the World Climate Research Programme (WCRP), co-sponsored by the World Meteorological Organisation (WMO), the Intergovernmental Oceanographic Commission (IOC) of UNESCO, and the International Science Council (ISC), set up the Grand Challenge on Near Term Climate Prediction, to make the case for, and understand the challenges in establishing routine operational climate predictions on these time scales (Kushnir et al., 2019). As of now, there are multiple ensembles of model hindcasts/re-forecasts available. These terms are used interchangeably in literature. In this study, for consistency, we will use the term re-forecast, defined here as a retrospective prediction (Boer et al., 2016), realized with the aim to evaluate them against observations and provide insight into our capacity of providing real decadal forecasts.

With the advent of operational predictions (Hermanson et al., 2022), there is growing research activity into the application of decadal forecasts (Dunstone et al., 2022). Glacier modeling is one such field, where there is a gap between seasonal modeling of glacier mass balance and runoff (e.g. Koziol and Arnold, 2018; Réveillet et al., 2018), and the more established modeling on the century and millennial scale, often using downscaled general circulation model (GCM) output (e.g. Huston et al., 2021; Rounce et al., 2023). By quantifying and improving the predictability of glacier mass balance on the decadal scale it may be possible to bridge this gap. If so, the resulting mass balance predictions could also be translated into glacier runoff, serving as an important input for water resource decisions, which often operate on this time scale (Kiem and Verdon-Kidd, 2011).

The aim of this study is to investigate the utility of forcing a mass balance model with decadal scale re-forecasts, to complement current mass balance modeling studies and the time scales they are commonly conducted on - centuries and millennia. As far as we know, this study is the first of its kind in large scale (regional to global) glacier modelling, following suit to testing the applicability of decadal re-forecasts in impact models for other research disciplines, such as marine biology (Payne et al., 2022) and the agricultural sector (Solaraju-Murali et al., 2022). A compacted review of applications, including a preliminary version of the current study, is presented in O'Kane et al. (2023). The current manuscript presents a modeling study using the Open Global Glacier Model (OGGM; Maussion et al., 2019), structured into two main components, on different spatial scales. The first component focuses on a set of 279 reference glaciers, while the second examines all global land-terminating glaciers. The OGGM simulations are driven by a multi-model, multi-member ensemble of monthly temperature and precipitation re-forecasts from the Coupled Model Intercomparison Project Phase 6 (CMIP6) Decadal Climate Prediction Project (DCPP; Boer et al., 2016). To evaluate the added skill of decadal re-forecast forcing, we compare these results with simulations using two alternative experiments: forcing with a simple persistence method, as well as with uninitialized, free-running historical GCM outputs and projections from the same models. This latter approach represents the traditional forcing method typically used for 21$^{\text{st}}$ century glacier simulations.

## 2    Data and Methods

### 2.1    Model

We use OGGM v.1.5.3 (Maussion et al., 2019, 2022) for the first component of the study – the simulation of reference glaciers – and a slightly updated version, in terms of calibration, for the global runs, which was not an official release (see Sect. 2.2.2). OGGM is an open-source modeling framework written in Python. It was developed to provide a catchment to global-scale, modular numerical modeling framework for various study set-ups of glacier evolution on multiple scales, while accounting for glacier geometry and ice dynamics. Maussion et al. (2019) and the continuously expanding and adapting model documentation at http://docs.oggm.org explain the model in detail and can be referred to for an in-depth model description. The current study focuses on the application of OGGM's mass balance modeling capabilities rather than the glacier flow.

#### 2.1.1    Methodology

The model takes a glacier-centric approach, using the outline of a glacier as a starting point. By default, the glacier outlines are automatically taken from the Randolph Glacier Inventory (RGI) version 6.0 (Pfeffer et al., 2014; RGI Consortium, 2017). Given a glacier outline and topographical and climate data, OGGM aims to: "i) provide a local map of the glacier including topography and hypsometry, (ii) estimate the glacier's total ice volume and compute a map of the bedrock topography, (iii) compute the surface climatic mass balance and (if applicable) its frontal ablation, (iv) simulate the glacier's dynamical evolution under various climate forcings and (v) provide an estimate of the uncertainties associated with the modeling chain" (Maussion et al., 2019; Recinos et al., 2019). In this study, we solely use the pre-processing and mass balance modeling capabilities of

the model, not the dynamical modeling tools. We call this a "fixed geometry approach", i.e. the surface area and elevation of the glacier are fixed when computing glacier wide mass balance. This simplification assumes that geometry feedbacks are negligible at annual-to-decadal scales and is justified by the dominance of other uncertainties, such as the unknown glacier state in the past (e.g. Eis et al., 2021). Furthermore, we only evaluate the climatic mass balance component and do not evaluate calving or other mass loss processes: from now on, we will use the term "mass balance" in place of "climatic mass balance" for simplicity.

### 2.1.2 Mass Balance

The mass balance module selected for the model set-up is the OGGM default in 1.5.3. Mass balance is calculated using a temperature index model which generates monthly accumulation and ablation along the glacier:

$$m_i(z) = p_f P_i^{\text{solid}} - \mu^* \max\left(T_i(z) - T_{\text{melt}}, 0\right) + \epsilon, \tag{1}$$

in which $m_i$ is monthly mass balance at elevation $z$, $T_i$ constitutes the monthly temperature that is adjusted based on its elevation by using a temperature lapse rate of 6.5 K km$^{-1}$. The threshold temperature $T_i(z) - T_{\text{melt}}$, above which melting occurs, is set to the default of -1 °C. Here, the precipitation correction factor $p_f$ is set to 2.5 globally (Maussion et al., 2019). $P_i^{\text{solid}}$ is the monthly solid precipitation, computed as a fraction of total precipitation, based on the monthly mean temperature. The temperature sensitivity of a glacier is indicated by calibrated parameter $\mu^*$, and $\epsilon$ is an optional residual, determined during calibration. In this study we make use of two independent calibration procedures, one making use of observations of glaciers with a mass balance record of at least five consecutive years (Sect. 2.2.1) and the other being based on a global dataset (Sect. 2.2.2).

### 2.1.3 Study Structure and Simulations

The study is divided into two components, each with different spatial scales, see Table 1. For the first component, we focus on 279 glaciers with direct mass balance observations from the World Glacier Monitoring Service (WGMS, N = 279 glaciers; WGMS, 2022). These glaciers, referred to as reference glaciers, represent land-terminating glaciers with robust observational records, spanning at least 5 consecutive years per glacier. Over the period 2000–2020, 2676 separate annual mass balance measurements are available for the 279 glaciers. For the second component, the mass balance of the approximately 214,000 land-terminating glaciers globally is simulated. In both components, the geometry of the glacier is based on the state at the RGI inventory date, usually between the years 2000 and 2010, and remains unchanged throughout the simulation. Validation with observed data is done for the period 2000–2020.

The study aim is to analyze forcing with decadal re-forecasts. However, in order to put the results into context, we perform a total of three experiments, with different complexity, in each component. The experiments represent forecasting from very simple to complex methods.

These experiments are defined as follows:

**1) Decadal re-forecast:** OGGM is forced with a 21-member multi-model ensemble of CMIP6 DCPP-A decadal re-forecasts (see Sect. 2.4 and Table 1). All different realizations (ensemble members) are downscaled to the glacier scale and run for all available decades.

*Final output*: a multi-model ensemble mean of results, from averaging results of the simulations with 21 members. This yields a time series with one mass balance value per year and glacier, 2000–2020.

**2) Persistence:** OGGM is forced with a simple, persistence-type forecast, where each period (lead times 1 to 9 years, see Sect. 2.3) is the same as the one that precedes it (from here on referred to as 'persistence forecast'). For this, we use baseline climate CRU. For example, the mass balance results from CRU forcing 1990–2000 form the persistence forecast for the period 2000–2010. Persistence forecasts are a typical null hypothesis against which other forecast skill is measured (Hargreaves, 2010).

*Final output*: a time series with one mass balance value per year and glacier, 2000–2020, based on the mean simulated mass balance under the baseline climate for the respective preceding decadade.

**3) GCM Historical:** OGGM is forced with a 21-member ensemble of CMIP6 historical simulations for 2000–2014, using climate projections for 2014–2020, see Sect. 2.4. Historical GCM runs and GCM projections are the current state of the art in forcing glacier models for the 20[th] and 21[st] century, including on the near-term time scale (see Hock et al., 2019, Slangen et al., 2017 and Zekollari et al., 2022 for overview papers).

*Final output*: a multi-model ensemble mean of results, from averaging results of the simulations with 21 members. This yields a time series with one mass balance value per year and glacier, 2000–2020.

The purpose of this combination of experiments is to assess the added value of the decadal re-forecasts over a naïve forecast method (persistence) and current state of the art (GCM historical), as well as analyze forecast skill of decadal and persistence (re-)forecasts at different lead times.

## 2.2 Mass Balance Model Calibration

For each component of our study, we carry out a separate model calibration. It must be noted that for the purpose of our study, the performance of the mass-balance model itself is secondary, since only the change in performance when using various forcing products is investigated. The calibrated parameters are held constant for each forcing product, allowing us to assess the impact of the forcing strategy alone, not the impact of calibration.

### 2.2.1 Component 1 - 279 reference Glaciers - Calibration with WGMS Data

For the first component, the mass balance calibration procedure is carried out with baseline climate CRU, see Sect. 2.4, over the years with observed data that fall within the CRU climate time series (1901–2020). We use the default calibration procedure as of OGGM 1.5.3, described in Marzeion et al. (2012) and Maussion et al. (2019). For all years with observations, the model output is then compared to observations, to identify the best candidate for $\mu^*$ and $\epsilon$ on a glacier-by-glacier basis. Because all

of these 279 glaciers have observations, the parameters do not need to be transferred to glaciers without observations and the mass balance model is calibrated to match observations over the calibration period.

### 2.2.2 Component 2 - Global Glaciers - Calibration with Geodetic Data

For the second component of this study, OGGM is calibrated separately, also using baseline climate CRU (Sect. 2.4). We benefit from the dataset by Hugonnet et al. (2021), providing geodetic mass balance estimates for 94 % of all global glaciers over the period 2000–2020. This dataset facilitates a broader calibration for the global runs, incorporating glaciers beyond the WGMS reference set. The monthly mass balance is computed as in Eq. (1) but without making use of the residual $\epsilon$, since this dataset consists of data for each glacier, removing the need for parameter transfer to glaciers without observations. More detailed information on the use of residual $\epsilon$ can be found in Marzeion et al. (2012). The calibration prioritizes temperature sensitivity parameter $\mu^*$, calibrated to match the glacier's geodetic mass balance of the period 2000–2020 Hugonnet et al. (2021). Note that the re-calibration for the global run is a practical necessity (we are simulating all glaciers globally) but has no bearing for our results, since we compare OGGM results with different forcing strategies, i.e. we are not assessing the model or its parameters but the forcing data used for the simulations. It must be added that in our study, we will always run the model during the period it has been calibrated for. This means that when run with the baseline climate CRU, it provides 'perfect results': exactly matching observations over the calibration period (bias of zero).

### 2.3 Lead Time and Ensembles

Due to the importance of the initial state of the climate system (atmosphere and ocean) for decadal prediction, forecast skill often declines with lead time (Zhu et al., 2019). Lead time here adheres to the definition by the American Meteorological Society: "*The length of time between the issuance of a forecast and the occurrence of the phenomena that were predicted*". To assess the lead time based skill in the context of our study, we create lead time based ensemble means of results in the decadal re-forecast experiment, to validate against observations. This results in nine time series of mass balances, 2000–2020, with input from lead times 1–9, respectively. Due to the clipping to hydrological years to match the WGMS measurements, lead time 0 does not exist in component 1. So for a decadal re-forecast initialized in 1990 (always in November), the first full year of simulated values is for the year 1992. In component 2, the first full year of simulated values would be 1991.

For the persistence experiment, we also create lead time based time series. Here, lead time refers to the forecast length, which is the same time period until the start of the forecasts. An example of lead time 2 persistence forecast would be the forecast period 2000–2001 being the same as the year 1998–1999. In the case of our study period 2000–2020, the lead time 1 persistence forecast uses temperature and precipitation from the time period 1999–2019, and lead time 9 therefore uses values from 1991–2011 for the forecast.

In our decadal re-forecast experiment, we create ensemble means of results as they would be utilized in practice. As Risbey et al. (2021) note, many assessments of re-forecast skill are likely overestimated, as the re-forecasts are informed by observations over the period assessed that would not be available to real forecasts. In order to avoid this as much as possible, we only assess a period that was not used in the drift correction (see Sect. 2.5) and use only lead times that would be available at the

beginning of the forecast period. This results in time series for each glacier and each full decade in the period 2000–2020. For an example decade, say 2000–2010, the ensemble mean of results consists of information from all re-forecasts initialized in 1990–2000. This means the ensemble size decreases over time, with only lead time 9 information being available in 2009. De-

creasing ensemble size at longer lead times typically increases forecast uncertainty, potentially affecting skill metrics slightly. We use information from all lead times available to maximize ensemble size, and in turn skill (Kadow et al., 2017). This approach again re-iterates the fact that decadal re-forecasts are multi-annual, rather than strictly ten years. In order to compare persistence as it would be used in practice, for decadal mean and cumulative mass balance forecasting, persistence forecasts at lead time 9 are applied.

**2.4 Climate Data**

The mass balance model in OGGM requires monthly climate in the form of reference height (2 m) air temperature and precipitation time series. The default baseline climate forcing, which we use for our persistence experiment, is the gridded Climatic Research Unit Time Series (CRU TS v4.01; Harris et al., 2020). This coarse (0.5°) dataset is then interpolated to a higher resolution climatology (CRU CL v2.0 at 10' resolution; New et al., 2002) following the anomaly mapping approach described

in Harris et al. (2020), to acquire climate time series with elevation data, which is not an attribute in the CRU TS. For each glacier, the monthly time series of temperature and precipitation are taken from the gridpoint closest to the glacier. Temperature is converted using an elevation-based lapse rate of 6.5 K km$^{-1}$ and precipitation is corrected using the default correction factor of $p_f$ = 2.5 (Maussion et al., 2019). While the CRU dataset constitutes our OGGM baseline climate for calibration, OGGM can also be supplied with GCM output that is bias corrected to the baseline climate, as we do in the decadal re-forecast and

GCM historical experiments.

In the decadal re-forecast experiment, OGGM is driven with a multi-model, multi-member retrospective ensemble of monthly temperature and precipitation re-forecasts from the DCPP component A, which provides re-forecasts. We use decadal realizations from the 'Flexible Global Ocean-Atmosphere-Land System Model' (FGOALS; Zhou et al., 2018), the 'Norwegian Climate Prediction Model' (NorCPM; Counillon et al., 2016; Bethke et al., 2021) and the 'Model for Interdisciplinary Research

on Climate version 6' (MIROC6; Tatebe et al., 2019; Kataoka et al., 2020). We use the r1i1p1f1-r1i1p1f10 realization of all models, where available. The DCPP-A decadal re-forecasts are initialized each year in the period 1960–2010, the first forecast year being 1961. The processing of the decadal data is explained below, in Sect. 2.5.

In the GCM historical experiment, we drive OGGM with temperature and precipitation from the historical iteration and projections of the same three GCMs, obtained from CMIP6 archived model output: FGOALS, NorCPM and MIROC6 (see

Table 1). The end of the historical simulation is in 2014 and data from 2015–2020 is provided by projection runs, leading to 11 full decades in the period 2000–2020. As the amount of available data becomes larger because of the different shared socio-economic pathways (SSP), the choice is to select certain SSPs or to introduce a discrepancy in ensemble size, if using all CMIP6 SSPs. We realize that neither option facilitates perfect comparison with the decadal re-forecast and persistence experiment. However, the benefit of comparison with projections outweighs these concerns, as initialized forecast vs. projections represents

the most realistic future use case. To preserve ensemble size, SSP245 was chosen as the projection for comparison, as it

**Table 1.** Graphical interpretation of the model set-up, including both components and experiments therein

| Component 1: WGMS Glaciers | | Component 2: Global Glaciers | |
|---|---|---|---|
| **Experiment** | **Climate Data Source** | **Model** | **Ensemble Size** |
| **Decadal Re-forecast** | CMIP6 DCPP-A: | FGOALS, NorCPM, MIROC6 | 21 |
| **Persistence** | - | CRU TS4.01 | 1 |
| **GCM Historical** | CMIP6 Historical and Projection runs | FGOALS, NorCPM, MIROC6 | 21 |

represents a medium pathway of future greenhouse gas emissions. From FGOALS runs, not enough SSP245 realizations were available at the time of the study, so SSP126, SSP245 and SSP585 time series were used. As these scenarios do not diverge significantly during 2015–2020, we still consider these results comparable.

In the pre-processing for this experiment, the time series are bias-corrected and downscaled to the glacier scale using a
variation of the delta method (Ramírez Villegas and Jarvis, 2010). Here, we take GCM anomalies relative to the 1961–1990 GCM mean for temperature and apply these to the CRU TS 4.01 (Harris et al., 2020) 1961–1990 means. The correction is applied monthly and ensures that mean and standard deviation are preserved during the bias correction period for temperature. Precipitation is corrected with a multiplicative factor and preserves only the mean. This is the standard method of processing GCM data for projection studies, (e.g. Zekollari et al., 2020), and is the reason we use it as an evaluation procedure for the
decadal re-forecasts.

## 2.5 Re-Forecast Drift Correction and Downscaling

In order to drive OGGM with the re-forecasts, each member is downscaled to the glacier scale using a statistical method applied with baseline climate CRU (New et al., 2002; Harris et al., 2020). Decadal re-forecasts experience a bias referred to as drift, because they start from an initialized state constrained by observations, which is inconsistent with the model's dynamics
(Kharin et al., 2012; Manzanas, 2020). The drift is lead time dependent because with time progressing, the model drifts away more from the initial state, towards a state more consistent with the model's climatology, which can lead to significant error (Pasternack et al., 2021). The re-forecasts have to be bias corrected to counter this error. Our correction adheres to recommendations in Boer et al. (2016), who recommend an overarching bias correction method, regardless of the initialization type of the forecast. The reasons behind these recommendations are discussed in-depth in e.g. Boer et al. (2016); Kharin et al.
(2012) and Hossain et al. (2022).

We assume that the bias contained in each member is model dependent and lead time dependent. Because of the assumption that the bias is different at and dependent on each lead time, subtracting a mean drift per member would lead to overcompensation at some lead times, and residual drift at others. For this reason, we create lead-time based climatologies per model, meaning one climatology over 1971–2000 which contains all lead time 1 years from the re-forecasts, one which con-
tains all lead time 2 years, and so on. These are then used to create anomalies relative to the baseline climate. For each model, each member is bias corrected according to:

$$d_t = \overline{T'_t} - \overline{CRU_{cl}}, \tag{2}$$

$$T'_{m,t,y} = T_{m,t,y} - d_t \tag{3}$$

In which $d_t$ is the average model bias or drift at each lead time $t$, calculated relative to the baseline climate input CRU. $CRU_{cl}$ is the CRU monthly climatology averaged over 1971-2000 (Harris et al., 2020; New et al., 2002). $T'_t$ is the model climatology at lead time $t$ calculated by averaging the lead time $t$ re-forecasts of all ensemble members of one model falling into the period 1971-2000. $T_{m,t,y}$ is the raw re-forecast, of member $m$ at lead time $t$ and for year $y$. $T'_{m,t,y}$ is the bias corrected re-forecast of member $m$ at lead time $t$ and for year $y$.

Lead time dependent bias correction is a fundamental step used in decadal forecasting, as is mean bias correction in future projections for impact modeling. Because the aim of our study is to compare the standard methodologies commonly applied in the respective fields, the GCM Historical data are processed as they would typically be in glacier modeling (e.g. Zekollari et al. (2024); Rounce et al. (2023) and many more), and the re-forecasts are processed according to the recommendations mentioned above. It must be added that the drift correction following lead time (the length of time between the issuance of a forecast and the occurrence of the phenomena that were predicted) does not apply to the GCM Historical experiment where we have only one simulation per realization.

## 3   Results and Discussion

This section evaluates the results of the three experiments - decadal re-forecast, persistence, and GCM historical - by comparing them against observed glacier mass balance data for individual years and decadal mean and cumulative mass balance. The mean and cumulative decadal mass balances are distinct from each other as the cumulative mass balances only include the simulated years for which observations exist. Throughout our analysis, we compare observational time series which inherently contain inter-annual variability to ensemble means from multi-member forecasts. It is important to recognize that ensemble means naturally reduce variability by averaging across multiple realizations, thereby smoothing internal fluctuations that occur in any single realization or observed series. This reduction in variability means that some portion of the absolute errors (e.g., the mean absolute error) used in the evaluation arises from the difference in variability between a single observational realization and the ensemble mean. Despite this limitation, ensemble means represent the most commonly used forecast product in practical applications due to their stability and reliability.

### 3.1   Component 1: WGMS Glaciers

We first focus on individual years, assessing how skill changes with lead time. To quantify model skill, we calculate the mean absolute error (MAE) and the Pearson correlation coefficient (r) for each lead time based ensemble mean, with the results

displayed in Fig. 1. The decadal re-forecast experiment shows consistent performance, with an MAE of 0.64 m w.e. at lead time 1 and a slightly lower value of 0.63 m w.e. at lead time 4, before increasing slightly to 0.65 m w.e. by lead time 9. Pearson correlation coefficients decrease from 0.14 at lead time 1 to 0.07 at lead time 9, reflecting the inherent difficulty of simulating individual annual mass balance values. For the persistence experiment, MAE increases from 0.69 m w.e. at lead time 1 to 0.72 m w.e. at lead time 9, with greater variability in correlation compared to the decadal re-forecast forcing. By contrast, results

from the GCM historical experiment, which does not depend on lead time, show constant skill. As there is no application of lead time to the GCM Historical experiment, its results (MAE = 0.66 m w.e., r = 0.01) are plotted as a constant, to compare the skill score magnitude. Overall, skill is low when simulating individual years, which is expected, and in line with other studies using OGGM for this set of reference glaciers. One example is Eis et al. (2021), who yield an MAE of 0.60 m w.e. with baseline climate CRU, over the time period from 1917 until the RGI outline date per glacier (often early 2000s). Simulating

single year mass balances is not the focus of this study, however, the results show that already on the single year level, forcing OGGM with decadal re-forecasts outperforms the persistence and the GCM historical experiments, which both show higher errors and lower correlations.

Next, we compare mean and cumulative mass balance over full decades, as illustrated in Fig. 2 and Table 2. To quantify model skill, we look at the mean error (ME), which is the average difference between observed and simulated values (sometimes

called "mean bias"), the mean absolute error (MAE) and the Pearson Correlation Coefficient. In the time period 2000–2020, we have 11 full decades for which the mean and cumulative mass balance are calculated per glacier. The full decades, which have significant overlap, are still compared separately and depicted in Figure 2 to show how the choice of decade can considerably impact skill statistics. For example, in the decadal re-forecast experiment, the decade 2001-2011 has a mean model error of -0.022 m w.e., whereas the decade 2002-2012 has a mean model error of 0.11 m w.e.

Observed mean mass balance for the decade 2000–2010 is -0.79 m w.e. Simulations using reanalysis data (CRU), which provides a benchmark for comparison, yield a mean error (ME) of -0.037 m w.e. and an MAE of 0.23 m w.e. Decadal re-forecasts produce comparable results, with an ME of 0.091 m w.e. and an MAE of 0.27 m w.e., while persistence forecasts display a larger ME of -0.16 m w.e. and an MAE of 0.39 m w.e. The GCM historical experiment shows slightly better performance than persistence, with an ME of -0.0059 m w.e. and an MAE of 0.29 m w.e. Correlation coefficients are moderate to high, with

decadal re-forecasts achieving the highest correlation (r = 0.64), followed by GCM historical (r = 0.61) and persistence (r = 0.58).

In terms of cumulative decadal mass balance, error patterns are similar. Decadal re-forecasts achieve an ME of -0.39 m w.e. and an MAE of 1.33 m w.e., while persistence experiments have an ME of -0.96 m w.e. and an MAE of 1.62 m w.e., reflecting the difficulty of using persistence forecasts for warming-sensitive variables like glacier mass balance. GCM historical

simulations show comparable results to decadal re-forecasts, with an ME of -0.27 m w.e. and an MAE of 1.58 m w.e.

To gauge the statistical significance of the differences between experiments, we carry out a two-tailed t-test (significance level 0.05) on the set of individual glaciers. The samples for these tests are the 279 glacier-specific values of decadal mean and cumulative mass balance for each forcing type. For decadal re-forecasts and GCM Historical experiments, these values are

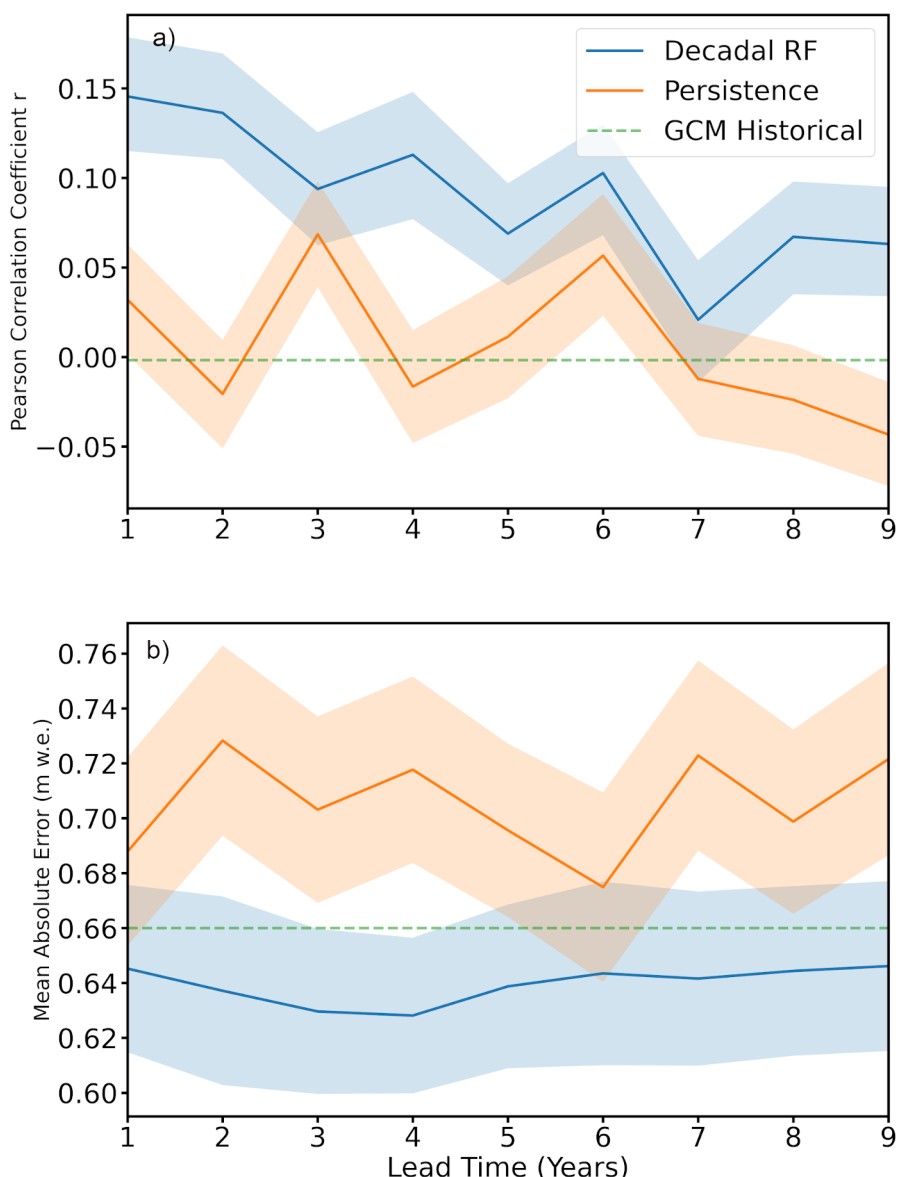

**Figure 1.** Forecast skill for annual mean mass balance for WGMS reference glaciers (N = 279). Forecast skill is given as the Pearson correlation coefficient (r) in a) and mean absolute error (MAE, m w.e.) between observed and simulated mass balance observations (N = 2676) in b). Skill is plotted as a function of lead time into the future, calculated across the appropriate comparison periods (2000–2020) for the decadal re-forecasts and persistence forecasts. The mean skill scores of the GCM historical (2000–2020) simulations also shown for reference, as a constant, since there is no application of lead time. Shaded areas for both persistence and decadal re-forecast (RF) denote the 90 % confidence interval estimated by bootstrapping: 5 % of the distribution is therefore above and 5 % below the shaded areas.

ensemble means, whereas for the persistence forecast, it is a single-member simulation. For the decadal mean mass balance, the difference between the decadal re-forecast and persistence experiment is statistically significant, as is the difference between the persistence and GCM Historical experiment. However, there is no significant difference between the decadal re-forecast and GCM Historical experiments. For the cumulative mean mass balance, there are no statistically significant differences amongst the experiments.

We also perform a binomial test, assessing improved skill by specifically evaluating reductions in mean absolute error for the decadal re-forecast relative to persistence or historical forcing. This shows that out of the 279 glaciers we analyze, 174 showed improved skill using decadal re-forecasts for decadal mean mass balance, and 186 showed improved skill for cumulative mass balance. Using a binomial significance, this suggests that the overall improvement from using decadal re-forecasts is significant at the 5 % level. We note that the t-test and binomial test can yield different interpretations because they test different aspects of statistical significance. The t-test compares the mean differences between forecast methods across the entire set of glaciers, assessing whether differences in the average performance are statistically distinguishable given the spread in the dataset. By contrast, the binomial test assesses significance based only on the count of glaciers showing improved performance (in terms of lower mean absolute error) when comparing one forecast method directly against another, irrespective of the magnitude of improvement.

Thus, while in some cases, the t-test does not indicate a statistically significant difference, due to variability across glaciers and the relatively small magnitude of improvement on average, the binomial test can indicate statistical significance simply because more glaciers improve under one approach compared to another than would be expected by chance alone. Both tests provide complementary perspectives: the t-test assesses the robustness of the average improvement magnitude across glaciers, whereas the binomial test evaluates consistency in improvement across individual glaciers.

The persistence experiment shows a notable overestimation of mass balance, particularly at longer lead times. This is evident in Fig. 2, where persistence forecasts systematically deviate from the observed values due to a lag in warming trends. This highlights the limitations of persistence-based methods for forecasting glacier mass balance on decadal time scales, where warming trends have significant influence, as is the case with glacier mass balance.

The skill differences between the experiments are further quantified in Table 2. Here for the reference glaciers, forcing with decadal re-forecasts outperforms forcing with persistence forecasts or historical GCM data, even though the absolute improvements in skill are small and statistically not significant between the decadal re-forecast and GCM Historical experiments, for individual glaciers. Pearson correlation coefficients between simulated and observed values are generally in the same range for all three experiments, with highest correlations for the decadal re-forecast experiment. All Pearson correlations are moderate for mean decadal mass balance - up to 0.64 for the decadal re-forecast experiment - and high - up to 0.85 - for cumulative decadal mass balance. Comparing these correlations to the low degree of correlation when simulating individual years emphasizes how the skill of our experiments lies primarily in simulating multi-annual averaged or cumulative mass balances, filtering out inter-annual noise. This is in line with similar approaches, where skill is found through integrating of fluxes over time, such as in seasonal snow accumulation (Förster et al., 2018).

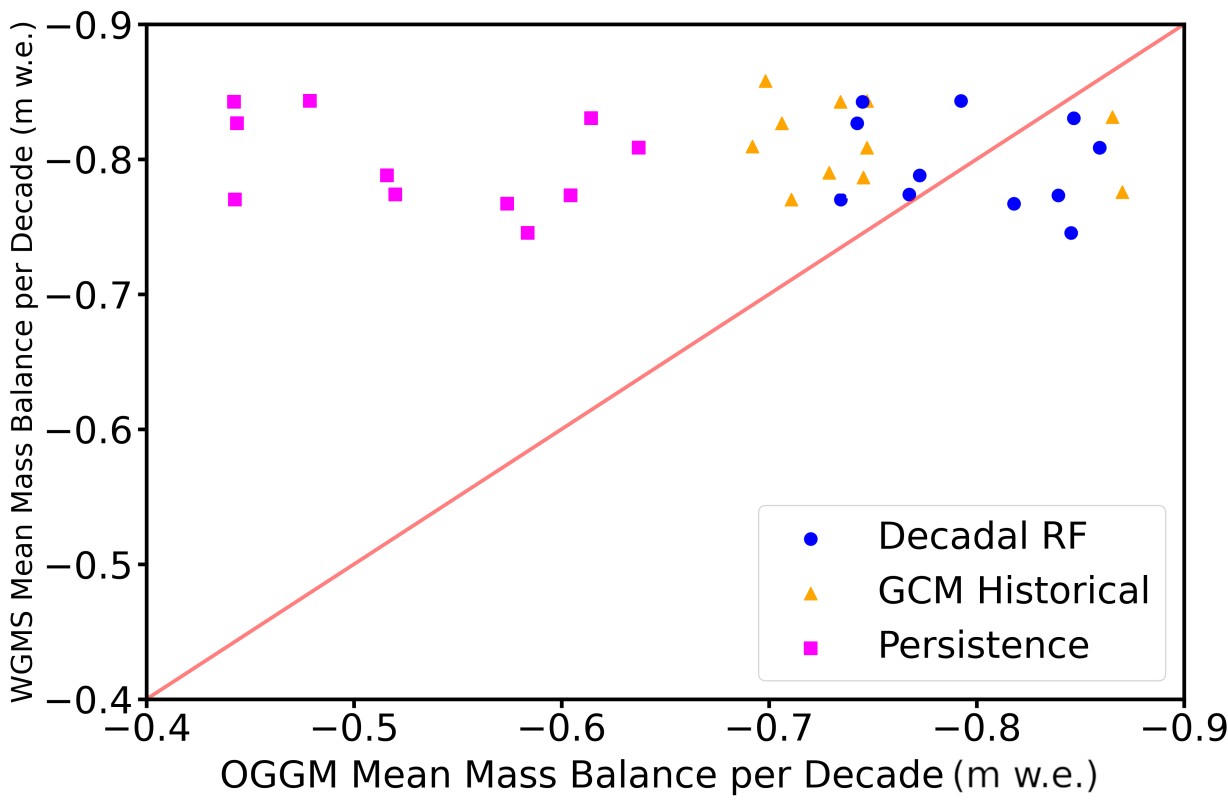

**Figure 2.** Simulated and observed mean mass balances over each full decade (N = 11), over all 279 reference glaciers. Only simulated values where observed values are available are used to generate the decadal means.

**Table 2.** Summary of the comparison between WGMS observed and simulated mass balances for all three experiments. The statistics shown are ME: model error, MAE: mean absolute error and Pearson correlation, as well as half the interquartile range of the particular statistic.

| Skill Measure | Decadal RF | Persistence | GCM Historical |
|---|---|---|---|
| Decadal Mean Mass Balance | | | |
| ME (m w.e.) | 0.091 ± 0.15 | -0.16 ± 0.20 | -0.0059 ± 0.21 |
| MAE (m w.e.) | 0.27 ± 0.16 | 0.39 ± 0.23 | 0.29 ± 0.22 |
| Pearson $r$ | 0.64 ± 0.09 | 0.58 ± 0.08 | 0.61 ± 0.18 |
| Decadal Cumulative Mass Balance | | | |
| ME (m w.e.) | -0.39 ± 0.20 | -0.96 ± 0.23 | -0.27 ± 0.26 |
| MAE (m w.e.) | 1.33 ± 0.66 | 1.62 ± 0.77 | 1.58 ± 0.62 |
| Pearson $r$ | 0.85 ± 0.12 | 0.74 ± 0.09 | 0.79 ± 0.09 |

We also note here that in the decadal re-forecast experiment, the multi-model ensemble mean of results yields higher skill than single-model ensembles. Comparing the multi-model ensemble mean of results to observations of mean decadal mass balance vs. single model ensemble means of results (N = 3) to observations, yields a decrease in MAE of 11 % (vs. FGOALS), 8 % (vs. NorCPM) and 6 % (vs. MIROC6), respectively. This aligns with our expectations from the literature, due to an increase in ensemble size and associated error cancellation and the separate forecast systems adding signal to the multi-model ensemble (Kadow et al., 2017; Delgado-Torres et al., 2022). Other studies applying multi-model ensembles in impact models, such as Payne et al. (2022), also come to the conclusion that a multi-model ensemble generally gives the best performance, which is why we do not explore single-model ensemble performance further.

## 3.2 Component 2: Global Glaciers

For all global land-terminating glaciers, we compare results of mean decadal mass balance in all experiments. To assess skill, 2000–2010 and 2010–2020 results are validated against the global geodetic mass balance dataset by Hugonnet et al. (2021). Because the time period 2000–2020 is covered by both validation datasets, a separate comparison of the two is provided in Sect. 3.3.

Comparing 2000–2010 decadal re-forecast skill scores to the persistence and GCM historical experiments, improvement in skill is slight (Table 3). Over the decade 2000–2010, decadal re-forecasts yield a 23 % and 18 % reduction in mean absolute error (MAE) relative to persistence and GCM historical forcing, respectively. Over this period, the MAE for the decadal re-forecast experiment was 0.28 m w.e., compared to 0.35 m w.e. for persistence forecasts and 0.33 m w.e. for the GCM historical experiment. The Pearson correlation coefficients similarly favor re-forecasts, potentially indicating higher overall skill on a global scale. Performing a two-tailed t-test (significance level 0.05) on the collection of glaciers shows that while the difference between the persistence experiment and the two other experiments is significant, the difference between the decadal re-forecast and GCM historical experiments is not.

Between the different glaciated regions of the world - RGI regions, see Fig. 3, there is considerable variation in skill. Indicated in the histograms in Fig. 4 are the regional mean absolute errors for 2000-2010, which vary considerably. The decadal re-forecast experiment achieves good agreement (regional mean difference ≤ 0.1 m w.e.) in 10 of 18 regions, reasonable agreement (difference 0.1–0.3 m w.e.) in seven regions, and mediocre agreement (difference ≥ 0.3 m w.e.) in one region. Variability in MAE is larger for the persistence and GCM historical experiments, with notable overestimation of mass balance due to lagging warming trends in persistence forecasts. The persistence experiment generally performs worse for regions sensitive to rapid warming, such as Greenland and Iceland, highlighting the limitations of static forecasts in a warming climate. Overall, the differences between experiments all lie within one standard deviation from the mean, for the simulations as well as within the mean error of observations (Hugonnet et al., 2021). All 2000-2010 observed values and their uncertainty, as well as all simulated values per experiment are included in Table 3.

The 2010-2020 results are slightly different than for the 2000-2010 period, in that the persistence experiment outperforms the decadal re-forecast experiment in overall skill score. The mean absolute error for the decadal re-forecast experiment is 0.28,

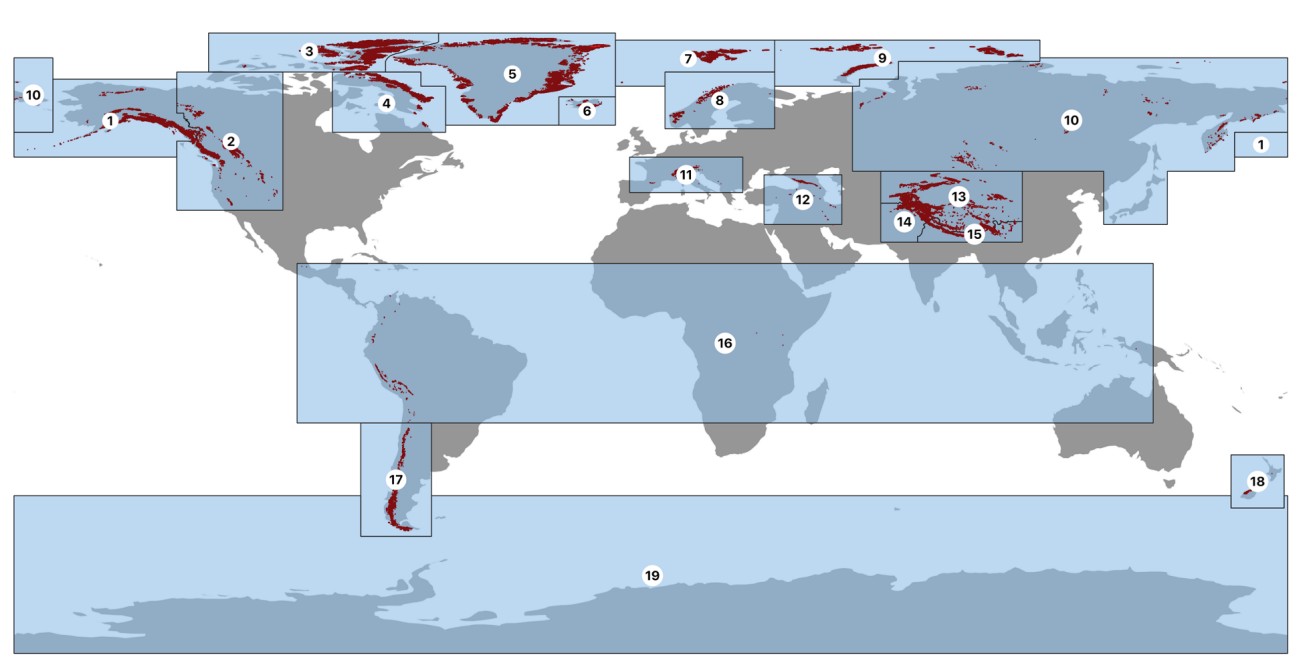

**Figure 3.** Map of all RGI regions, of which we use 18 in OGGM as we exclude Antarctica (region 19). Numbered light blue polygons correspond to the glacier regions (GTN-G, 2023) listed in Table 4 and Table 5. Dark red areas correspond to all glaciers listed in the RGI inventory (RGI Consortium, 2017). The country outlines are made with Natural Earth country polygons (https://www.naturalearthdata.com, last access: 02 January 2024).

while persistence mean absolute error is 0.24, with Pearson correlation coefficients of 0.69 and 0.77 respectively. In terms of goodness of fit per region however, the decadal re-forecast experiment slightly outperforms the persistence experiment, with 11 of 18 regions showing good fit (defined as a difference between regional means =< 0.1 m w.e.). Five of 18 regions show reasonable fit (difference between regional means 0.1 – 0.3 m w.e.) and two regions show mediocre fit (difference between regional means >= 0.3 m w.e.). The persistence experiment shows 8 regions with good fit, 8 regions with reasonable fit and two regions with mediocre fit. The exact goodness of fit numbers, including the observations and observational uncertainty can be found in Table 5.

The fact that the persistence experiment performs markedly better for this decade than for the previous one, while skill scores and goodness of fit are similar for the decadal re-forecast experiment, lies in the calibration. As explained in Sect. 2.2.2, the calibration and validation periods are the same. Because of the nature of persistence forecasts, the forcing data for the 2000-2010 period originated from 1990–2000, and was not used in calibration. For the period 2010-2020, however, both the climate data for the persistence forecast (2000-2010) and the forecasted decade were part of the calibration, resulting in a bias

**Table 3.** Observed and simulated global mass balance skill scores. Summary of the comparison between global observed and simulated mean mass balances for all three experiments in the decades 2000-2010 and 2010-2020. The statistics shown are ME: model error, MAE: mean absolute error and Pearson correlation, as well as half the interquartile range for the particular statistic.

| Skill Measure | Decadal RF | Persistence | GCM Historical |
|---|---|---|---|
| 2000-2010 | | | |
| ME (m w.e.) | $0.082 \pm 0.15$ | $0.24 \pm 0.14$ | $0.18 \pm 0.17$ |
| MAE (m w.e.) | $0.28 \pm 0.12$ | $0.35 \pm 0.26$ | $0.33 \pm 0.11$ |
| Pearson $r$ | $0.71 \pm 0.09$ | $0.64 \pm 0.08$ | $0.65 \pm 0.09$ |
| 2010-2020 | | | |
| ME (m w.e.) | $-0.043 \pm 0.17$ | $-0.098 \pm 0.18$ | $-0.069 \pm 0.14$ |
| MAE (m w.e.) | $0.28 \pm 0.14$ | $0.24 \pm 0.12$ | $0.31 \pm 0.16$ |
| Pearson $r$ | $0.69 \pm 0.08$ | $0.77 \pm 0.09$ | $0.66 \pm 0.10$ |

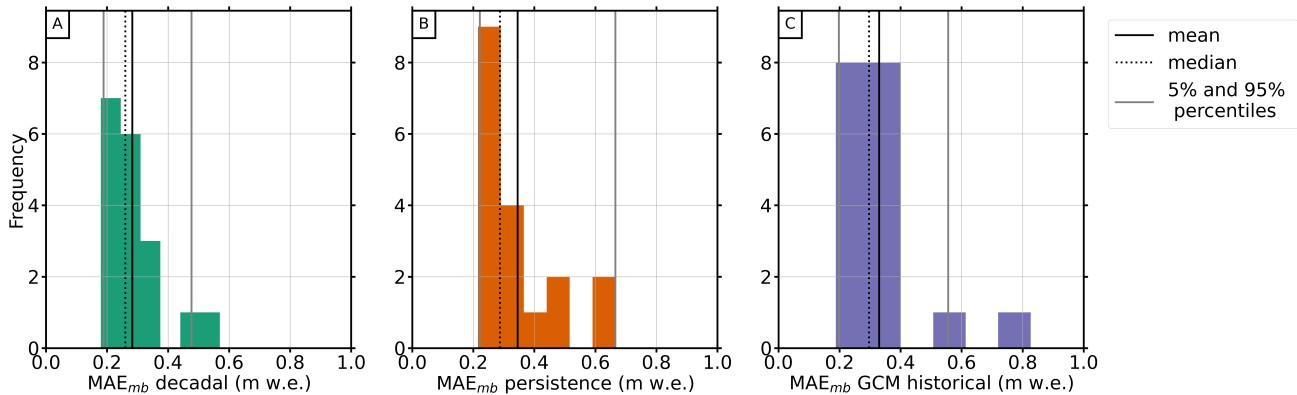

**Figure 4.** Mean absolute error for runs with decadal re-forecast forcing (a), persistence forcing (b) and GCM historical forcing (c), for all RGI regions (N = 18). The vertical lines indicate mean (bold, black), median (dotted, black) and 5th and 95th percentiles (grey) of the mean absolute error.

of 0 for the full time period 2000–2020. To assess the effect of this, we run another persistence simulation, this time with a model only calibrated for the period 2000-2010, leaving 2010-2020 for validation. This is also the scenario that would occur when using persistence to forecast 2020-2030: the model is calibrated for the decade prior to the forecasted one. This leads to a markedly worse score for the persistence forecast, with a mean absolute error of $0.38 \pm 0.36$ m w.e. and a Pearson correlation coefficient of 0.26, indicating low correlation. For the sake of consistency, keeping calibration the same for all experiments, the persistence experiment results presented in Table 5 are from the simulation with the original calibration.

**Table 4.** Color-coded overview of mass balances over the period 2000–2010, from the decadal re-forecast, persistence and GCM historical experiments, as well as observed data from Hugonnet et al. (2021). The Hugonnet et al. (2021) values include their uncertainties. The table is color coded according to goodness of fit between experiment and Hugonnet et al. (2021) data: cells in bold indicate a good fit (difference between regional means =< 0.1 m w.e.), cells in normal font a reasonable fit (difference between regional mean 0.1-0.3 m w.e.) and cells in italics indicate mediocre fit (difference between regional mean >=0.3 m w.e.).

| RGI Region | Mean mass balance 2000-2010 (m w.e.) | | |
| --- | --- | --- | --- |
| | Observed (Hugonnet et al., 2021) | Decadal re-forecast | Persistence | GCM Historical |
| 1 Alaska | -0.29 ± 0.47 | **-0.34** | **-0.28** | -0.0055 |
| 2 Western Canada / US | -0.18 ± 0.48 | *-0.42* | **-0.10** | **-0.077** |
| 3 Arctic Canada North | -0.35 ± 0.28 | -0.11 | -0.062 | *-0.0024* |
| 4 Arctic Canada South | -0.40 ± 0.37 | **-0.32** | -0.13 | -0.16 |
| 5 Greenland Periphery | -0.34 ± 0.38 | **-0.28** | *0.31* | *0.10* |
| 6 Iceland | -0.42 ± 0.35 | -0.57 | *-0.056* | -0.22 |
| 7 Svalbard and Jan Mayen | -0.26 ± 0.28 | **-0.18** | -0.0089 | -0.10 |
| 8 Scandinavia | -0.46 ± 0.45 | -0.22 | *0.044* | **-0.41** |
| 9 Russian Arctic | -0.31 ± 0.26 | **-0.23** | -0.083 | -0.13 |
| 10 North Asia | -0.38 ± 0.58 | **-0.42** | -0.27 | -0.27 |
| 11 Central Europe | -0.60 ± 0.62 | -0.36 | *0.040* | *0.21* |
| 12 Caucasus/ Middle East | -0.35 ± 0.50 | **-0.35** | -0.070 | **-0.27** |
| 13 Central Asia | -0.21 ± 0.46 | **-0.16** | -0.054 | **-0.11** |
| 14 South Asia West | -0.08 ± 0.48 | **-0.09** | 0.048 | **-0.033** |
| 15 South Asia East | -0.34 ± 0.48 | -0.13 | **-0.25** | **-0.30** |
| 16 Low Latitudes | -0.33 ± 0.49 | **-0.34** | **-0.41** | **-0.37** |
| 17 Southern Andes | -0.17 ± 0.56 | -0.09 | **-0.17** | -0.053 |
| 18 New Zealand | -0.060 ± 0.61 | -0.15 | 0.13 | -0.18 |
| Total | -0.31 ± 0.45 | **-0.26** | *-0.076* | -0.13 |

**Table 5.** Color-coded overview of mass balances over the period 2010–2020, from the decadal re-forecast, persistence and GCM historical/ projection experiments, as well as observed data from Hugonnet et al. (2021). The Hugonnet et al. (2021) values include their uncertainties. The table is color coded according to goodness of fit between experiment and Hugonnet et al. (2021) data: cells in bold indicate a good fit (difference between regional means =< 0.1 m w.e.), cells in regular font a reasonable fit (difference between regional mean 0.1-0.3 m w.e.) and cells in italics indicate mediocre fit (difference between regional mean >=0.3 m w.e.).

| RGI Region | Mean mass balance 2010-2020 (m w.e.) | | | |
| --- | --- | --- | --- | --- |
| | Observed (Hugonnet et al., 2021) | Decadal re-forecast | Persistence | GCM Historical and Projection |
| 1 Alaska | -0.57 ± 0.45 | **-0.51** | -0.32 | -0.34 |
| 2 Western Canada / US | -0.50 ± 0.51 | -0.58 | -0.23 | **-0.45** |
| 3 Arctic Canada North | -0.40 ± 0.27 | -0.21 | **-0.36** | -0.23 |
| 4 Arctic Canada South | -0.45 ± 0.35 | -0.58 | **-0.37** | **-0.51** |
| 5 Greenland Periphery | -0.22 ± 0.36 | -0.41 | -0.38 | -0.37 |
| 6 Iceland | -0.25 ± 0.33 | -0.56 | -0.43 | -0.51 |
| 7 Svalbard and Jan Mayen | -0.30 ± 0.27 | **-0.32** | **-0.23** | **-0.22** |
| 8 Scandinavia | -0.36 ± 0.44 | **-0.46** | **-0.40** | -0.58 |
| 9 Russian Arctic | -0.29 ± 0.23 | **-0.39** | **-0.29** | **-0.26** |
| 10 North Asia | -0.44 ± 0.59 | **-0.53** | **-0.39** | **-0.50** |
| 11 Central Europe | -0.59 ± 0.63 | **-0.58** | -0.39 | **-0.54** |
| 12 Caucasus/ Middle East | -0.58 ± 0.54 | **-0.58** | *-0.27* | -0.76 |
| 13 Central Asia | -0.27 ± 0.47 | **-0.28** | -0.16 | **-0.32** |
| 14 South Asia West | -0.14 ± 0.47 | **-0.046** | **-0.08** | **-0.23** |
| 15 South Asia East | -0.49 ± 0.49 | *0.11* | -0.38 | **-0.52** |
| 16 Low Latitudes | -0.32 ± 0.49 | *-0.69* | **-0.25** | *-0.96* |
| 17 Southern Andes | -0.28 ± 0.34 | **-0.28** | -0.022 | **-0.31** |
| 18 New Zealand | -0.34 ± 0.65 | **-0.41** | *-0.03* | -0.45 |
| Total | -0.38 ± 0.44 | **-0.42** | **-0.29** | **-0.45** |

### 3.3 Observed Data Differences

Finally, this study clearly benefits from the availability of two separate data sets of observed mass balance for the same 2000–2020 time period. This not only allows for more critical assessment of model results, but also of uncertainty within observations. The use of both data sets here warrants a comparison between overlapping observations. For the period 2000-2010, we calculate the mean mass balance for all glaciers where the WGMS data set has full observations throughout the decade (N = 90). For this subset of 90 glaciers, we find a mean bias/difference of -0.049 m w.e. and an absolute bias/difference of 0.23 m w.e.

between the WGMS and geodetic mass balance data. For the period 2010-2020 we have 100 glaciers with uninterrupted mass balance coverage from the WGMS data set. Comparing to the Hugonnet et al. (2021) mean mass balances yields a mean bias/difference of -0.15 m w.e. and an absolute bias/difference of 0.32 m w.e.. For the full time period 2000–2020, a sub-set of 67 glacier has observations throughout the full time period. The mean bias/difference here is -0.11 m w.e. and the absolute bias/difference is 0.23 m w.e.. These results are along the same magnitude of error we observe when comparing our decadal

re-forecast simulation results to observations, as well as the uncertainties associated with the Hugonnet et al. (2021) data (see Tables 4 and 5). This reinforces the need for caution when interpreting observations but also confirms the satisfactory quality of the simulated results.

### 3.4 Skill

On the whole, the skill displayed in the decadal re-forecast experiment is comparable to or slightly better than in the other

experiments. Regional differences in skill ('goodness of fit' in the global component) in the re-forecast experiment likely stem from differences in re-forecast quality. This means the degree of correspondence between observed and simulated temperature and precipitation. We refer to Delgado-Torres et al. (2022) for a comprehensive analysis of the quality of the re-forecasts used here. Their results show generally high skill for DCPP forecasts of temperature, especially over land masses. For precipitation however, skill is limited in several regions, including central Europe (region 11) and Western Canada/ US (region 2), which

also show the least mass balance skill out of all regions (see Table 4 and Table 5). Good precipitation skill is observed for northern Europe and Central Asia, in line with our yielded 'good fit' results for Svalbard and Jan Mayen (region 7) and Central Asia (region 13) (Tables 4 and 5). Delgado-Torres et al. (2022) find the quality of decadal re-forecasts of temperature higher than historical temperature simulations for multiple regions, while added value is smaller when simulating temperature. With accurate representation of precipitation being essential for mass balance modeling, it is likely that precipitation forecasts

are a limiting factor in seeing significant improvement when simulating near-term mass balance with decadal re-forecasts as forcing, over persistence or historical/projection data. Additionally, addressing precipitation in a more complex calibration and downscaling approach may decrease uncertainty in this area. This is to be kept in mind when designing future studies, especially if these include regions where predictive skill for precipitation is low.

The primary source of decadal (re-)forecast skill, beyond free-running simulations, is initialization. The main benefit of

420 initialized decadal forecasts is that the initialization allows them to capture both the response to external forcing and the phase of the internal variability of the climate system. For example, initialized re-forecasts better capture Atlantic multi-

decadal variability than GCM historical simulations, in a study by García-Serrano et al. (2015). Smith et al. (2019) note that improvements from initialization generally take place in regions where the uninitialized simulations already have some skill. This can also be seen in the similar skill patterns between the GCM Historical and decadal re-forecast experiments in Tables 4 and 5. Initialized forecasts better represent internal climate variability due to accurate initial conditions (Smith et al., 2019). This enhances their predictive skill for time scales of 1–10 years compared to uninitialized projections. Initialized decadal forecasts also provide a more constant uncertainty over time than uninitialized projects, whose uncertainty increases significantly with lead time Strobach and Bel (2017). Future improvements in initialization, e.g. regarding the initialization of the North Atlantic Oscillation (Nicolì et al., 2025), may therefore still offer potential for reducing uncertainties in near-term glacier modeling, even if the current benefits are limited.

In addition, decadal forecasts outperform uninitialized projections in representing regional climate variability, especially in temperature and precipitation, which are crucial for accurately modeling glacier mass balance. As studies such as Thornton et al. (2014) describe, climate variability often exacerbates the impact of climate change on vulnerable communities. In a glacier context, this could mean e.g. above-average melt events impacting downstream communities, so accurate prediction is essential. Finally, Payne et al. (2022) note in their assessment of forcing with decadal re-forecasts vs. uninitialized projections that there is an established demand for communicating both likely values and uncertainty of a forecast made with an impact model (Bruno Soares and Dessai, 2016). When the effort is to minimize this uncertainty and make a forecast as precise as possible, forcing with initialized forecasts is likely preferable on the decadal scale. Despite these advantages, decadal forecasts are still in development, and the continuation of projects such as the DCPP contribution to CMIP6 (Boer et al., 2016) is essential to ensure their operational use.

This study also reveals arguments for applying decadal (re-)forecasts over persistence or uninitialized projections in future near-term glacier modeling studies. Especially warming and glacier mass loss accelerating (Hugonnet et al., 2021) is an argument for forcing near-term simulations with initialized forecasts over persistence forecasts, which lack the warming trend over the simulation period. The choice of initialized forecast forcing over uninitialized projections stems mainly from the probabilistic context: projections, especially as pathway differences increase over time, are inherently more uncertain than a forecast initialized at the beginning of the forecast period. Statistically, we observe no significant difference between the decadal re-forecast and GCM Historical experiment per individual glacier (section 3.1). Binomial tests however show a general improvement when forcing OGGM with decadal re-forecasts. Also, Smith et al. (2019) note that significance tests, such as applied here, often underestimate the improvement from initialization because there is a significant overlap of skill in both experiments (e.g. simulating the global warming trend). Smith et al. (2019): *"This common signal introduces a bias that is not taken into account in standard significance tests and diminishes their power (Siegert et al., 2017)."* Therefore, the small improvements in the skill statistics of the decadal re-forecast experiment over the other experiments may indicate a larger benefit of forcing glacier models with decadal re-forecasts than is evident in the Student's t-test. This poses that even modest skill increases from initialization may be reliably attributed to improved forecast initialization rather than random chance.

## 4   Conclusion and Outlook


Our results show that there is merit in using decadal scale forecasts in glacier modeling, as they show good predictive skill of averaged multi-annual mass balances. We see that, indicated by lower errors and higher correlations, the use of decadal re-forecasts yields comparable or better results than forcing OGGM with a persistence forecast or the current state of the art: GCM historical data of temperature and precipitation. Forcing OGGM with decadal re-forecasts, a binomial test shows improvement for a majority of the WGMS glaciers and globally, we see good or reasonable agreement between simulated and observed mean mass balances for almost all RGI regions, and on a glacier-to-glacier basis. Both forcing with GCM historical/projection simulations and decadal re-forecasts yields skillful predictions of cumulative mass balance over single decades for the WGMS set of reference glaciers, providing an important basis for modeling the amount of mass moving downstream over a decade. Planning future studies with these forcings of course operates on the assumption that real time decadal forecasts (for decades that lie in the future) and GCM projections would be of similar quality to the re-forecasts and historical runs used in the current study, and would benefit from future validation. The use of decadal forecasts would not replace GCM projections for $21^{st}$ century glacier modeling, but can provide added clarity on the near-term, especially in terms of uncertainty.



The results shown here are limited by multiple factors and we especially highlight the need for continuing this research with a larger ensemble, which could increase predictive skill (Smith et al., 2013). We also propose a more detailed look into ensemble spread, explicitly quantifying and incorporating ensemble spread as a measure of uncertainty. Probabilistic evaluation methods, such as the Continuous Ranked Probability Score (CRPS; Hersbach (2000)), could better utilize the full predictive distribution provided by ensemble forecasts, thus giving a more comprehensive picture of forecast uncertainty and reliability. Another important step towards applications in hydrology and industry would be the use of decadal forecasts to force OGGM dynamically, as opposed to the static mass balance in the current study. This would mainly serve to ensure a more accurate initial state of the glacier, important for areas where glaciers have already changed significantly since their RGI inventory date, such as the European Alps.



Finally, the foremost aim when continuing this research is to have the highest possible quality near-term glacier simulations for the next decade. Accurate knowledge of near-term trends is essential, as these time scales are most relevant for applications in hydrology and industry (Frans et al., 2016; Lane and Nienow, 2019; Arheimer et al., 2024), especially in regions where populations are directly affected in the form of water scarcity or flooding. This work would add to a growing database of field cases utilizing near-term forecasts, see e.g. O'Kane et al. (2023). Our results support the case for using decadal forecasts to achieve this, rather than depending only on the continuation or repetition of recent decadal climatic conditions (as in persistence forecasts). The next applications of the methods laid out in this study would be on basin- and global scales, forcing OGGM with a multi-model ensemble of decadal forecasts, into the 2030s. OGGM would be applied to acquire decadal estimates of future mean and cumulative mass balance, volume and area change as well as glacier runoff. Results could provide robust, important information on the amount of glacier mass lost and moving downstream in the form of runoff. With the continuing and



accelerating impacts of climate change on glaciers and water resources, we emphasize the need for these near-term predictions, in order to best inform and protect the communities dependent on them.

*Code and data availability.* All OGGM source code is available via https://github.com/OGGM, with documentation available at https://docs.
oggm.org/en/v1.5.3/. All decadal re-forecast and GCM data is available via https://esgf-node.llnl.gov/search/cmip6/. The OGGM baseline climate CRU is available in the OGGM pre-processed directories https://cluster.klima.uni-bremen.de/~oggm/gdirs/oggm_v1.4/exps/CRU_new/elev_bands/qc0/pcp2.5/match_geod_pergla.

## Appendix A: Supplement to van der Laan et al.: Decadal re-forecasts of glacier climatic mass balance

The two figures below have been produced as an example of results for a single glacier. As can be seen, neither the GCM
Historical nor the decadal re-forecast experiments perform very well when analyzed at the single glacier level, in this case the Hintereisferner and Langfjorjoekulen. In the Hintereisferner case, also cumulatively, neither the GCM Historical nor the Decadal re-forecasts have provided more skill than the very simple persistence experiment. The ensemble spread, in this case, is even larger than for the GCM Historical experiment, which does not speak for its benefits of initialization. In the Langfjord-joekulen case, the decadal re-forecast experiment performs better than the GCM historical and persistence experiments. It does
not follow the year-to-year variations, but captures the mean mass balance over the decade much better than the other two experiments. Overall, simulating single glaciers with OGGM is not the model's fortitude. The model is calibrated using the baseline climate and publicly available data, which is either sparse (WGMS) or only provides a decadal mean (geodetic global dataset, Hugonnet et al. (2021)). Thus, overall benefits of using a different climate forcing, such as in this study, should be determined over a larger set of glaciers.

*Author contributions.* LL helped conceive the study, co-developed the drift correction, did the analysis and wrote the paper. AV ran the global simulations and improved writing and general study design. AAS contributed to the study design and development of the drift correction. FM is the main developer of OGGM and contributed to study design, model set-up and improving the text. KF set up the original study concept and contributed to discussions and improving writing.

*Competing interests.* The authors declare that no competing interests are present.

*Acknowledgements.* LL, employed by the project Global glacier mass balance prediction on seasonal and decadal scale "GLISSADE", was funded by the Deutsche Forschungsgemeinschaft (DFG, German Research Foundation), Grant No. 416069075 (FO1269/1). AV was funded by the Austrian Science Fund (FWF) project P30256 and the German Research Foundation (DFG) – MA 6966/5-1. AAS was supported by

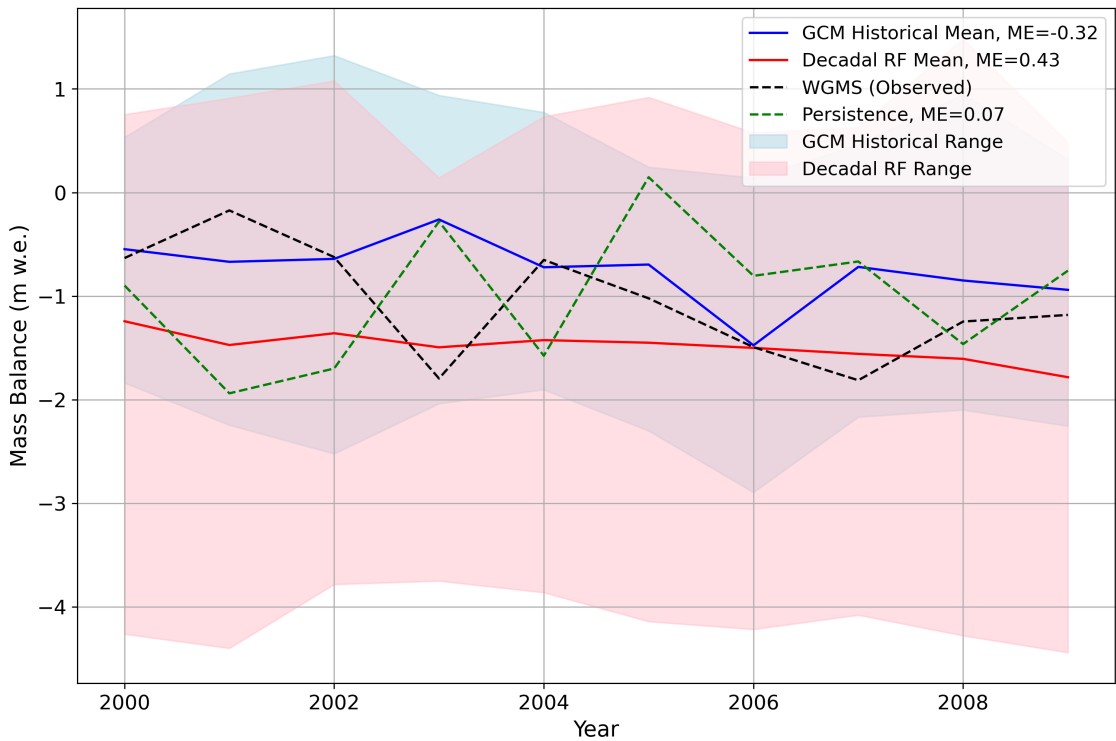

**Figure A1.** Single glacier result example for the Hintereisferner, Austria. Mean errors (ME) for the different experiments, for mean mass balance over the decade, as in Table 2, are indicated in the legend. This means the difference between the mean observed mass balance and the mean simulated mass balance for the different experiments. For the GCM historical and decadal RF experiments, 'simulated' refers to the ensemble mean.

the Met Office Hadley Centre Climate Programme (HCCP) funded by the UK Department for Science, Innovation and Technology (DSIT). The authors would like to thank Dr. Mark Payne for the helpful discussion on decadal forecasting and Dr. Lizz Ultee for her encouragement while completing this study.


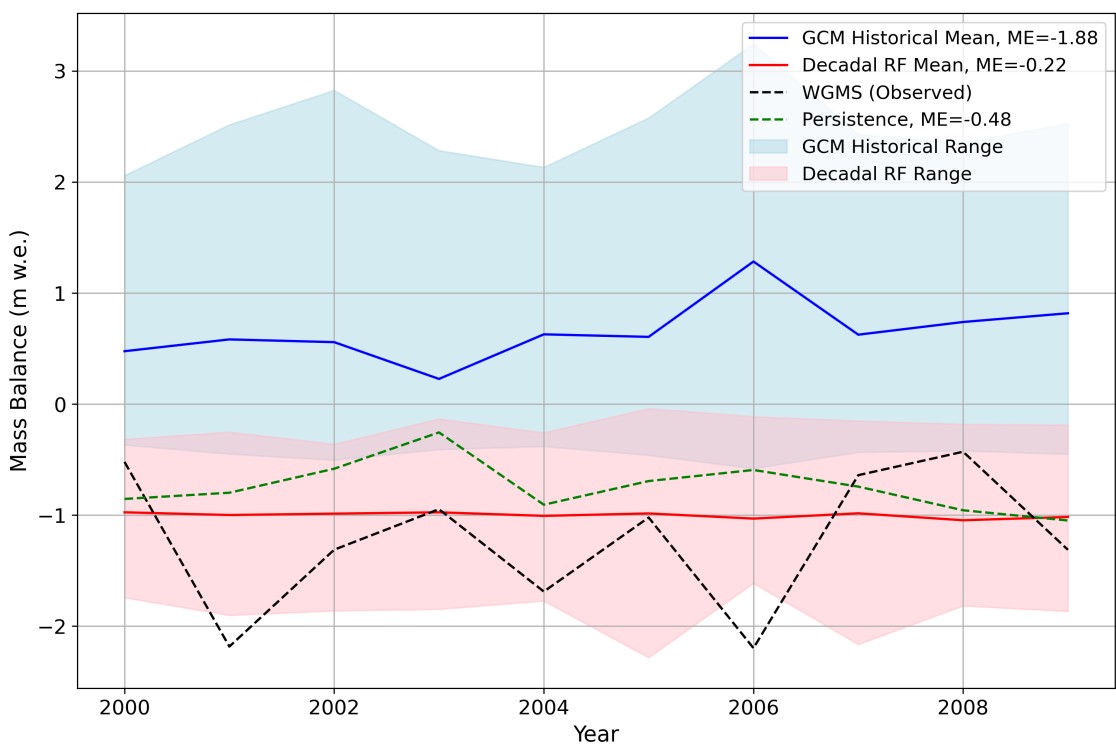

**Figure A2.** Single glacier result example for the Langfjordjoekulen, Norway. Mean errors (ME) for the different experiments, for mean mass balance over the decade, as in Table 2, are indicated in the legend. This means the difference between the mean observed mass balance and the mean simulated mass balance for the different experiments. For the GCM historical and decadal RF experiments, 'simulated' refers to the ensemble mean.

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
