# Peer review of "Decadal re-forecasts of glacier climatic mass balance"

_EGUsphere, 2024_

## Referee Comment (RC1)

**Review of "Decadal re-forecasts of glacier climatic mass balance" by *van der Laan et al*.**

This work aims to assess the applicability of forcing a glacier mass balance model with decadal re-forecasts that could help bridge the gap between seasonal and centurial-millennial timescales. The authors use the mass balance scheme of the Open Global Glacier Model and conduct three experiments (1) persistence runs using CRU forcing, (2) GCM historical runs with a 21-member CMIP6 ensemble, and (3) decadal re-forecast experiment using the same ensemble from the Decadal Climate Prediction Project (DCPP). They conclude that decadal re-forecasts have similar or better fit to the observed mass balance as compared to the other two experiments.

Overall, this is a detailed assessment with adequately designed experiments to address the study objective. I have two major comments regarding the clarity of the methods and the presentation of the results. The manuscript will benefit from a restructuring of the Methods section for better understanding of the experiment design and the specifics of each of the three experiments performed. Second, the Results and Discussion section requires improvement as it currently lacks rigor in the presentation of the statistical analysis and the discussion of the results.

I have separated the major comments for the Methods and Results/Discussion section below, followed by a few minor comments. Hopefully, this will help the authors to revise and resubmit the manuscript.

**1)      Major comments:**

**Methods:**

- Section 2.2: The application and purpose of the two calibration approaches requires better explanation. Why is there a need to calibrate with WGMS data for the 279 glaciers only and how does that feed into the experiments? Why not just use the calibration of ~214000 glaciers with the geodetic MB for the experiments?
- Why does one calibration approach have $\varepsilon$ and the other does not – are they both not done for individual glaciers?
- For 2.2.1, how are the two unknowns ($\mu^*$ and $\varepsilon$) established with one equation?
- Ln 115: What is the re-calibration step here that is done for the global run?
- Was the calibration with geodetic data also done with CRU (similar to the WGMS calibration)?
- Ln 116: Does this mean that once $\mu^*$ and $\varepsilon$ are established for each glacier (using CRU), the same values will be used for all experiments?
- 2.2.2 calibration was done over the 2000-2020 period, what about the 2.2.1 calibration?

- Section 2.2.3 needs restructuring for clarification of the experiment design. For example, information in Ln 123 – 130 can be merged with the individual experiment information. It seems Ln 123 – 126 is describing the persistence experiment?
- The manuscript talks about two components (e.g., Ln 123 and 127) and three experiments. It seems the two components refer to the two calibration approaches? Later, the results are separated for Reference and Global glaciers, and this was not clear in the objectives (Introduction) or the methods section. Ln 109 mentions about the 'global component of the study', but these components are never defined.
- The model run years require better explanation as well: the simulations are done over 1990 – 2020 period; Ln 117 states that "we will always run the model during the period it has been calibrated for"; Ln 112 states that the calibration with the geodetic MB is done for the 2000 – 2020 period; Ln 130 states that the validation is carried over the 2000 – 2020 period (calibration and validation here are supposedly used interchangeably?) – all this requires a clearer description.
- I understand the authors created the separate sections on Experiments (2.2.3), Lead Times (2.3), and Climate Data (2.4) for clarity, but it made following the methods somewhat cumbersome. I recommend merging all the information on the three experiments under the experiment design section, including what climate data was used and how the lead times were defined. And then, summarize this information in a table.

**Results and Discussion:**

- Ln 232: Where is this N = 2676 coming from? The WGMS calibration approach has N = 279 and I presume calibration with geodetic MB has N = 214,000 (Ln 129)? I think these first few lines should be in Methods rather than results.

  I am also confused regarding the Fig.1 caption mentioning N = 279 reference glaciers for getting the forecast skill and then in the next sentence stating N = 2676 for $r$ and MAE.

- Ln 235 – 245: I understand the authors want to share these results to highlight that year-to-year prediction is not practical with the current modelling scheme and is not the objective of the manuscript. But this paragraph is putting too much emphasis on the statistics without providing much context. For example, what does a MAE of 0.6 or 0.7 m w.e. mean, is this too high or too low? What are the annual mass balance magnitudes in general? Perhaps a metric like Mean Absolute Percentage Error (MAPE) will be more informative here.

  In general, I think the authors can remove this paragraph and Fig. 1 altogether and keep the focus on decadal timescales only (please see a minor comment as well regarding the definition of decadal vs annual timescales).

- Ln 254: This statement needs better qualification; how is a model error threshold of <0.2 m w.e. established? Is this statement referring to the first row (ME) of Table 2? I

suggest the authors establish more rigor in defining the statistical thresholds for good and bad results based on observed MB estimates and physical explanations. I am struggling to understand whether an error is too small, too large, or just right for the arbitrarily selected N = 279 (or 2676) glaciers.

- Ln 265: I am not sure I understand this correctly, the decadal forecast MAE (0.29 m w.e.) is 7% larger than GCM Historical MAE (0.27 m w.e.), but this sentence suggests there is a 7% reduction in decadal as compared to GCM forecast.

  Are these differences significant, not just statistically but also in general terms (e.g., would these differences affect global or regional scale assessments to understand glacial mass loss or melt rates, etc.). This is important because the GCM Historical forecast metrics are similar to the decadal re-forecast ones in most cases, sometimes with slightly better results as well.

  For this entire paragraph, it would be helpful to understand the meaning behind the mean and cumulative mass balances and the ME and MAE, and where these differences are coming from for these select glaciers.

- Ln 275: Which figure or table are these results referring to?

- Ln 295 onward: It would significantly improve the narrative if the authors were to dissect the regional differences and provide a better explanation of where the "considerable variation in skill" is coming from. These results (Fig. 4, Tables 4 and 5) are the more interesting part of this study but the presentation of the results and discussion here is somewhat deficient (the text repeats the statistics in the tables, but their meaning and importance is not explained).

- Ln 300: Earlier a threshold of <0.2 m w.e. was used for 'good' results. These thresholds should be consistent across the analysis (and perhaps specified earlier in the methods section on how the metrics were established).

  Also, the errors in the geodetic MB from *Hugonnet et al.* needs to be accounted for. For example, in Table 4, Region 10 has a MB of -0.38 ± 0.58. Why is -0.42 considered a good fit but -0.27 a reasonable fit based simply on the mean MB value?

- Ln 323: Where are these results shown? In fact, shouldn't these be the main results to ensure that the three experiments are comparable by design and the results are not affected by the calibration/validation periods.

**2)      Minor comments:**

Ln 23: *"...glaciers were the largest contributor to sea-level rise..."* Is this specifically referring to glaciers outside the polar regions, in continuation to Ln 20? Can you please cite this.

Ln 38: It is best to keep the terms consistent. It does not make sense to use "decadal prediction" or decadal timescales for single years or durations <10-years.

Ln 47: In applications of?

Ln 55: The common time scales here are referring to centuries and millennia?

Ln 57: What are "impact models"?

Ln 94: Can you please provide a justification for why the precipitation correction factor is set to 2.5 for all glaciers globally and for all forcing data sources? The *Maussion et al. 2019* citation alone is not adequate. Does this affect the MB computations for persistence experiments (using CRU) vs GCM historical or decadal RF experiments?

Ln 101: What is the first component of the study? This was mentioned earlier in Ln 67 as well which needs clarification. The last paragraph of the Introduction can benefit from explicit enumeration of the objectives and the "components" of the study.

Ln 106 – 107: Can you please clarify and rephrase this statement (on *"…parameters do not need to be transferred … and are therefore well constrained"*).

Ln 110: 94% of the RGIv6 glacier count?

Ln 133: "*All different realizations are downscaled to the glacier scale…*" What does this downscaling to glacier scale mean?

Ln 148: It is best to call it the persistence experiment only and not introduce a new term for this (i.e., naïve forecast).

Ln 240: What does remarkably consistent mean? These are just statistical results, so it is best not to use such superlatives.

Ln 258: Can you clarify what the ten-year lag of warming means.

Ln 289: Please rephrase "*slight but clearly noticeable*". In a tabular form, a difference in the third decimal place will also be clearly noticeable.

Is Fig. 4 for 2000 – 2010 period?

---

## Author Comment (AC1)

**Review of "Decadal re-forecasts of glacier climatic mass balance" by *van der Laan et al*.**

This work aims to assess the applicability of forcing a glacier mass balance model with decadal re-forecasts that could help bridge the gap between seasonal and centurialmillennial timescales. The authors use the mass balance scheme of the Open Global Glacier Model and conduct three experiments (1) persistence runs using CRU forcing, (2) GCM historical runs with a 21-member CMIP6 ensemble, and (3) decadal reforecast experiment using the same ensemble from the Decadal Climate Prediction Project (DCPP). They conclude that decadal re-forecasts have similar or better fit to the observed mass balance as compared to the other two experiments.

Overall, this is a detailed assessment with adequately designed experiments to address the study objective. I have two major comments regarding the clarity of the methods and the presentation of the results. The manuscript will benefit from a restructuring of the Methods section for better understanding of the experiment design and the specifics of each of the three experiments performed. Second, the Results and Discussion section requires improvement as it currently lacks rigor in the presentation of the statistical analysis and the discussion of the results.

I have separated the major comments for the Methods and Results/Discussion section below, followed by a few minor comments. Hopefully, this will help the authors to revise and resubmit the manuscript.

Dear editor and Reviewer 1,

Our sincere thanks for the thorough review and detailed comments. We will make several revisions to address the concerns raised. Below, we provide detailed responses to each of the comments. Our replies to the comments are in blue.

Thank you again for your time, kind regards,
Larissa van der Laan, on behalf of the author team

**1)    Major comments:**

**Methods:**

- Section 2.2: The application and purpose of the two calibration approaches requires better explanation. Why is there a need to calibrate with WGMS data for the 279 glaciers only and how does that feed into the experiments? Why not just use the calibration of ~214000 glaciers with the geodetic MB for the experiments?

We understand why this can be confusing. In part, this has to do with the trajectory of our experiments and the development of OGGM. We began this study with the WGMS glaciers only, and the OGGM default calibration as of v. 1.5.3, which is to

calibrate the mass balance model with the WGMS glaciers. The global analysis was added later, as the global geodetic dataset became available for calibration and validation. Using both approaches gives us additional insights: WGMS MB data is relatively sparse, though has a yearly resolution and has data preceding the global geodetic data, while the geodetic data has a lower temporal resolution, but a global coverage. The two datasets are also not perfect and do not necessarily agree at the glacier scale. Therefore, the calibration with WGMS data ensures that we have the best available model for each glacier individually, to focus on the impact of each forcing dataset, and then similarly for the global analysis.

- Why does one calibration approach have $\varepsilon$ and the other does not – are they both not done for individual glaciers?
- For 2.2.1, how are the two unknowns ($\mu^*$ and $\varepsilon$) established with one equation?

In older OGGM versions, $\varepsilon$ was introduced to apply a calibration where no WGMS data was available. For the purpose of our study, where we calibrate only for glaciers with data, $\varepsilon$ is always close to 0. We refer to Marzeion et al. (2012) and Maussion et al. (2019) Sect. 3.3. for an in-depth discussion of the calibration procedure, but it must be noted that for the purpose of our study, the performance of the mass-balance model itself is only secondary, since only the change in performance when using various forcing products is investigated.

- Ln 115: What is the re-calibration step here that is done for the global run?

In essence, the addition of another analysis, global this time. This was awkwardly worded and will be changed to: "Note that the separate calibration for the global run…"

- Was the calibration with geodetic data also done with CRU (similar to the WGMS calibration)?

Yes

- Ln 116: Does this mean that once $\mu^*$ and $\varepsilon$ are established for each glacier (using CRU), the same values will be used for all experiments?

Yes, the parameters are held constant for each forcing product, allowing us to assess the impact of the forcing strategy alone, not the impact of calibration.

- 2.2.2 calibration was done over the 2000-2020 period, what about the 2.2.1 calibration?

  The calibration here was done over the years with observed data for the WGMS glaciers that fall within the CRU climate data period (1901-2020).

- Section 2.2.3 needs restructuring for clarification of the experiment design. For example, information in Ln 123 – 130 can be merged with the individual experiment information. It seems Ln 123 – 126 is describing the persistence experiment?

- The manuscript talks about two components (e.g., Ln 123 and 127) and three experiments. It seems the two components refer to the two calibration approaches? Later, the results are separated for Reference and Global glaciers, and this was not clear in the objectives (Introduction) or the methods section. Ln 109 mentions about the 'global component of the study', but these components are never defined.
- The model run years require better explanation as well: the simulations are done over 1990 – 2020 period; Ln 117 states that "we will always run the model during the period it has been calibrated for"; Ln 112 states that the calibration with the geodetic MB is done for the 2000 – 2020 period; Ln 130 states that the validation is carried over the 2000 – 2020 period (calibration and validation here are supposedly used interchangeably?) – all this requires a clearer description.

We acknowledge the need for restructuring this section to improve clarity. The two components mentioned refer to the two sets of analyses/approaches — the reference glaciers and the global glaciers. The three experiments (persistence, GCM historical, and decadal re-forecast) were applied separately to these two components.

The manuscript will be revised to clearly define these components in the Introduction and Methods sections, with explicit mention of how they relate to the experimental design. Furthermore, the timeline for model runs, calibration, and validation periods will be clarified to avoid any confusion regarding the simulation years. The objectives in the Introduction will be updated to reflect these clarifications.

- I understand the authors created the separate sections on Experiments (2.2.3), Lead Times (2.3), and Climate Data (2.4) for clarity, but it made following the methods somewhat cumbersome. I recommend merging all the information on the three experiments under the experiment design section, including what climate data was used and how the lead times were defined. And then, summarize this information in a table.

Thank you for this suggestion. We tried various solutions for restructuring but concluded that the original structure still suits the logical flow best. We will however amend the text, making it more concise and hopefully more understandable.

**Results and Discussion:**

- Ln 232: Where is this N = 2676 coming from? The WGMS calibration approach has N = 279 and I presume calibration with geodetic MB has N = 214,000 (Ln 129)? I think these first few lines should be in Methods rather than results.

  I am also confused regarding the Fig.1 caption mentioning N = 279 reference glaciers for getting the forecast skill and then in the next sentence stating N = 2676 for $r$ and MAE.

The number N = 2676 refers to the total number of annual observations across the 279 WGMS reference glaciers, not the number of glaciers themselves. This distinction was not clearly communicated.

We will move the explanation of N = 2676 to the Methods section and clarify that this number represents the total number of annual observations rather than the number of glaciers.

- Ln 235 – 245: I understand the authors want to share these results to highlight that year-to-year prediction is not practical with the current modelling scheme and is not the objective of the manuscript. But this paragraph is putting too much emphasis on the statistics without providing much context. For example, what does a MAE of 0.6 or 0.7 m w.e. mean, is this too high or too low? What are the annual mass balance magnitudes in general? Perhaps a metric like Mean Absolute Percentage Error (MAPE) will be more informative here.

  In general, I think the authors can remove this paragraph and Fig. 1 altogether and keep the focus on decadal timescales only (please see a minor comment as well regarding the definition of decadal vs annual timescales).

  We understand the concern that the emphasis on single-year predictions may detract from the main focus on decadal timescales. However, we do think this section provides important context on lead time and its importance in decadal-scale forecasting. In accordance with comments by reviewer 2, we have also added background information on lead time and decadal scale forecasting in general. We are unsure about how to apply MAPE in the context of mass-balance, which can have positive, zero, or negative values. We will change the text however to emphasize the relative differences between the various skill values, and discuss these values in the context of observational uncertainty for context.

- Ln 254: This statement needs better qualification; how is a model error threshold of <0.2 m w.e. established? Is this statement referring to the first row (ME) of Table 2? I suggest the authors establish more rigor in defining the statistical thresholds for good and bad results based on observed MB estimates and physical explanations. I am struggling to understand whether an error is too small, too large, or just right for the arbitrarily selected N = 279 (or 2676) glaciers.

We agree that this needs more rigor. The 279 glaciers, however, are not selected arbitrarily, they are the WGMS glaciers with observations over a time period longer than 5 consecutive years and are land-terminating. This information will now be included in the manuscript.
The <0.2 m w.e. threshold is indeed arbitrary and will be removed.

- Ln 265: I am not sure I understand this correctly, the decadal forecast MAE (0.29 m w.e.) is 7% larger than GCM Historical MAE (0.27 m w.e.), but this sentence suggests there is a 7% reduction in decadal as compared to GCM forecast.

Thank you for pointing out this error! This was mixed up in the table and should be 0.27 m w.e. For the decadal re-forecast experiment, not the GCM historical experiment.

Are these differences significant, not just statistically but also in general terms (e.g., would these differences affect global or regional scale assessments to understand glacial mass loss or melt rates, etc.). This is important because the GCM Historical forecast metrics are similar to the decadal re-forecast ones in most cases, sometimes with slightly better results as well.

For this entire paragraph, it would be helpful to understand the meaning behind the mean and cumulative mass balances and the ME and MAE, and where these differences are coming from for these select glaciers.

We hope that the following context may provide some clarity:

On average, the 279 WGMS glaciers lost 0.79 m w.e. during the decade 2000-2010. As a control against an ideal case, and to gauge the magnitude of errors, OGGM is also forced with reanalysis dataset CRU and run for the 279 WGMS glaciers. This data is also used in calibration and to create the persistence forecast. Comparing against observed WGMS data of mean mass balance per decade, the errors +- the half interquartile range are as follows:
Mean error  -0.037 +- 0.16, mean absolute error 0.23 +- 0.17 and the Pearson correlation 0.72 +- 0.11.

Comparing to the experiment errors in table 2, this means the errors for both the decadal re-forecast and GCM historical experiment are within the order of magnitude of the error when forcing OGGM with 'ideal' CRU data. The forcing with CRU reanalysis data does, as expected, lead to better results than forcing with a forecast or projection. For all glaciers, in a binomial test, results when forced with CRU are closer to observed mean mass balance than in the forcing experiments.

The order of magnitude however, gives confidence in the skill of either experiment's forcing. For cumulative mass balance, the error statistics for the decadal re-forecast and GCM historical experiment are also within 10% of the error when forcing OGGM with CRU data.

To gauge the significance of experiment differences from a statistical point of view, we carry out a two-tailed t-test (significance level 0.05). For the decadal mean mass balance, the difference between the Decadal RF and Persistence experiment is statistically significant, as is the difference between Persistence and GCM Historical experiment. However, there is no significant difference between Decadal RF and GCM Historical experiments. For the cumulative mean mass balance, there are no statistically significant differences between the experiments.

In accordance with our response to reviewer 2, this emphasizes that the improvement in skill is notable but not necessarily significant. We will amend the text to make sure that our conclusions reflect these results accurately.

- Ln 275: Which figure or table are these results referring to?

Multi-model ensemble results on average show higher skill than single-model realizations. We will refer to a citation in the manuscript to make it clear that this is not one of our results.

- Ln 295 onward: It would significantly improve the narrative if the authors were to dissect the regional differences and provide a better explanation of where the "considerable variation in skill" is coming from. These results (Fig. 4, Tables 4 and 5) are the more interesting part of this study but the presentation of the results and discussion here is somewhat deficient (the text repeats the statistics in the tables, but their meaning and importance is not explained).

We appreciate the suggestion to delve deeper into the regional differences. These variations are indeed significant and warrant a thorough discussion. However, as noted in a response to reviewer 2, we aim to discuss the added value of using decadal re-forecasts, rather than their inherent skill at simulating (regional) climate. We will make clear that we think the regional SMB differences stem from differences in skill predicting temperature and precipitation in the respective regions, and refer to the relevant literature discussing this skill and its sources.

- Ln 300: Earlier a threshold of <0.2 m w.e. was used for 'good' results. These thresholds should be consistent across the analysis (and perhaps specified earlier in the methods section on how the metrics were established).

The earlier (quite arbitrary) threshold has been removed, so the thresholds are consistent throughout the manuscript now.

Also, the errors in the geodetic MB from *Hugonnet et al.* needs to be accounted for. For example, in Table 4, Region 10 has a MB of -0.38 ± 0.58. Why is -0.42 considered a good fit but -0.27 a reasonable fit based simply on the mean MB value?

We will amend the color coding/ goodness of fit criteria to reflect the Hugonnet errors. Thank you for this idea.

- Ln 323: Where are these results shown? In fact, shouldn't these be the main results to ensure that the three experiments are comparable by design and the results are not affected by the calibration/validation periods.

We agree that these should be the results shown for the persistence experiment instead. This will be changed in the manuscript, with the explanation of the calibration period.

**2)    Minor comments:**

Ln 23: *"...glaciers were the largest contributor to sea-level rise..."* Is this specifically referring to glaciers outside the polar regions, in continuation to Ln 20? Can you please cite this.

The statement refers specifically to glaciers outside the ice sheets. We will clarify this.

Ln 38: It is best to keep the terms consistent. It does not make sense to use "decadal prediction" or decadal timescales for single years or durations <10-years.

This, although confusing at times, is necessary to remain consistent throughout the manuscript, since our decadal re-forecasts are e.g. clipped to hydrological years, hence not 10 full years. It is only clarified so explicitly here to avoid confusion later on.

Ln 47: In applications of?

Thank you for spotting this error. This should read "[...] into the application of decadal forecasts."

Ln 55: The common time scales here are referring to centuries and millennia?

Yes, which we have also clarified.

Ln 57: What are "impact models"?

Impact models refer to models that assess the consequences of climate change on natural and human systems, such as glacier runoff or agricultural productivity (e.g. ISIMIP paper).

Ln 94: Can you please provide a justification for why the precipitation correction factor is set to 2.5 for all glaciers globally and for all forcing data sources? The *Maussion et al. 2019* citation alone is not adequate. Does this affect the MB computations for persistence experiments (using CRU) vs GCM historical or decadal RF experiments?

The precipitation factor is computed for historical data (here, CRU) by minimizing the error in variance of the mass-balance for all 279 WGMS glaciers. For another historical dataset (e.g. ERA5), the precipitation factor would be different indeed. The forcing climate datasets however (re-forecasts, historical GCMs, etc.) are then bias corrected to the historical data and therefore have a glacier specific correction depending on the bias correction method used for each product (re-forecasts or GCMs) according to practices commonly used in the large scale modeling literature.

Ln 101: What is the first component of the study? This was mentioned earlier in Ln 67 as well which needs clarification. The last paragraph of the Introduction can benefit from explicit enumeration of the objectives and the "components" of the study.

This will be clarified

Ln 106 – 107: Can you please clarify and rephrase this statement (on *"…parameters do not need to be transferred … and are therefore well constrained"*).

This was unclear indeed and refers to our reply to the comment above regarding the mu and epsilon parameter. Before the availability of global geodetic observations, parameters needed to be transferred to glaciers without any observation (Zekollari et al., 2024), leading to substantial errors. In our case, we apply the model to glaciers with either in-situ (WGMS) or geodetic observations, meaning that the MB model is calibrated to match observations. The statement "well constrained" however was not correct because of equifinality (e.g. Schuster et al., 2023). This sentence will be revised to convey the intended meaning: that the MB model is calibrated to match observations over the calibration period.

Ln 110: 94% of the RGIv6 glacier count?

Yes

Ln 133: "*All different realizations are downscaled to the glacier scale…*" What does this downscaling to glacier scale mean?

This is explained in section 2.4, and we will make sure the text references this section.

Ln 148: It is best to call it the persistence experiment only and not introduce a new term for this (i.e., naïve forecast).

We have included this term because it may be more familiar to readers and give added context to the term persistence.

Ln 240: What does remarkably consistent mean? These are just statistical results, so it is best not to use such superlatives.

We agree this can be misleading. We've revised the manuscript to avoid superlatives.

Ln 258: Can you clarify what the ten-year lag of warming means.

The "ten-year lag of warming" refers to the delay in temperature increases observed in persistence forecasts compared to actual observations.

Ln 289: Please rephrase "*slight but clearly noticeable*". In a tabular form, a difference in the third decimal place will also be clearly noticeable.

We will rephrase this

Is Fig. 4 for 2000 – 2010 period?

Yes

References:

Marzeion, B., Jarosch, A. H., & Hofer, M. (2012). Past and future sea-level change from the surface mass balance of glaciers. *The Cryosphere*, *6*(6), 1295-1322.

Schuster, L., Rounce, D. R., & Maussion, F. (2023). Glacier projections sensitivity to temperature-index model choices and calibration strategies. *Annals of Glaciology*, 1-16.

Zekollari, H., Huss, M., Schuster, L., Maussion, F., Rounce, D. R., Aguayo, R., ... & Farinotti, D. (2024). 21 st century global glacier evolution under CMIP6 scenarios and the role of glacier-specific observations. *EGUsphere*, *2024*, 1-33.

---

## Author Comment (AC2)

Author response to reviews of "Decadal re-forecasts of glacier climatic mass balance" by van der Laan et al.
(https://doi.org/10.5194/egusphere-2024-387)

Dear editor and reviewer 2,
We would like to thank you for the thorough review and detailed comments. This has been a great help in understanding where our manuscript needs improvement. We propose to make changes to the manuscript in accordance with our replies below, in blue.

Thank you again for your time, kind regards,

Larissa van der Laan, on behalf of the author team

This paper presents an analysis of glacier mass balance forecasts on multi-year to decade timeframes, using the mass-balance module of the Open Global Glacier Model. The authors compare forecasts made with climate forcing from Global climate model historical simulations, observation-initialized reforecasts from decadal prediction models, and a simple persistence forecast. Comparing their simulated mass balance re-forecasts to both in-situ and geodetic glacier mass balance observations, they conclude that using the initialized climate reforecasts provides an improvement in skill over GCM-based forcings. The predictability of glacier mass balance on short timeframes is an important problem, and the overall approach of comparing these forcing strategies using the mass balance model and assessing them against observations seems sound. However, there are some significant issues with the clarity of methods and results. It is not clear to me that the authors have demonstrated a meaningful difference in skill between reforecasts and GCM-based forcings. My main comments are detailed below, with some additional minor comments later on.

We thank the reviewer for this concise summary and evaluation of our work. We hope the answers below and the edits of our manuscript will improve the clarity.

Major comments:
1) Significance of skill improvements
The principal finding is that reforecasts provide an improvement in skill over using GCM output as forcing, and they emphasize the improvements for decadal mean and cumulative balance (since the skill for yearly forecasts is low; e.g., Fig 1 and line 244). However, there appears to be a huge spread in the Mean Absolute Error (MAE) metrics, such that the 1-sigma ranges substantially overlap. For example, in table 2 for decadal means, the authors report MAE of 0.29 +/- 0.32 mw.e. using reforecasts, and 0.27 +/- 0.31 mw.e. using GCMs. Overlaps are even greater for cumulative balance: 1.33 +/- 3.21 and 1.58 +/- 2.96. Presumably the standard deviations correspond to the distribution of errors across the individual WGMS glaciers. It's hard to see how this decrease in MAE is significant, given such wide distributions. The authors do not really comment on the wide spread of these error statistics. At the very least, they need to be discussed and the overall conclusions put in the context of these wide distributions.

We acknowledge the large spread in the error metrics and that this means it may be difficult to demonstrate significance for individual glaciers. To make the overall improvement more evident, we performed a binomial test. Out of the 279 glaciers we analyze, 174 showed improved skill using decadal re-forecasts for decadal mean mass

balance, and 186 showed improved skill for cumulative mass balance. Using a binomial significance, this suggests that the *overall* improvement is significant at the 5% level.

At a more technical level, there are some other issues with reported statistics that I find puzzling and not well explained.

We agree that the explanations for these points were lacking and will add clarifications accordingly.

To return to the MAE metric in table 2, the +/- range in many cases exceeds the central value reported, implying negative values, which don't make sense for Mean Absolute Error which should be positive definite. If the standard deviation of a positive-definite distribution is so large, does that imply a very long tail and some very large errors?

The large standard deviation indeed implies a long tail and very large errors here. Below is an MAE histogram example of the 2000-2010 decade from the GCM Historical experiment, which illustrates the spread of the errors. Especially the few very large errors (above 1 m w.e.), which are also present for the other experiments, affect the standard deviation. In the revised manuscript, we will change all statistics to use quantile ranges instead of mean and standard deviation to reflect that the distributions are not gaussian.

[Figure]

*Fig 1. Histogram of the MAE for the GCM Historical experiment 2000-2010*

Also in Table 2, what is the Model Error statistic? As far as I can tell this is never explained.

Thank you for alerting us to this. We use "Mean Error". We will add the following text to explain the Mean Error statistic: "To quantify model skill, we look at the mean error (ME),

which is the average difference between observed and simulated values (sometimes called "mean bias"), the mean absolute error (MAE) and the Pearson Correlation Coefficient."

At line 250 it is stated that the period 2000-2020 gives 11 full decades. This is technically true in a moving-window sense but these 11, 1-decade windows are not statistically independent samples. Why are these reported as different decades? There is a lot of potentially independent information across different individual glaciers, so why is Fig. 2 plotted in terms of these 11 heavily overlapping windows?

Together, these make it hard to interpret the significance of the overall conclusions.

It is true that there is significant overlap between the decades. Unfortunately, because of the limited availability of decadal re-forecasts, and wanting to assess the re-forecasts in the manner that future forecasts would be used (i.e. forecasts outside the bias correction time frame 1971-2000), we can only use the time period 2000-2020.
However, because within the relatively short window of a decade, a year's difference - or the choice of the specific decade, in other words - can significantly affect results. This is a hallmark of working on this time scale, but we agree this was not sufficiently explained in the manuscript. To illustrate this, we added the following to section 3.1:

*The full decades, which have significant overlap, are all compared separately and depicted in Figure 2 to show how the choice of decade can impact skill statistics. For example, in the Decadal Re-forecast experiment, the decade 2001-2011 has a mean model error of -0.022 m w.e., whereas the decade 2002-2012 has a mean model error of 0.11 m w.e.*

2) ReForecast drift correction
I found the explanation of the reforecast bias correction to be confusing. The authors note that the bias correction is lead-time dependent, but do not really explain why (lines 215-16). This would seem to be an important point to explain thoroughly in order to compare GCM to reforecast-based glacier simulations.

This is now explained better and we refer to relevant papers for more in-depth insights and limit our explanation to the following:

Decadal re-forecasts experience a bias which grows with lead time and is referred to as drift (Kharin et al., 2012; Manzanas et al., 2020). The drift is lead time dependent because the model drifts away from the initial state as the prediction progresses, towards a state more consistent with the model's climatology, which can lead to significant error (Pasternack et al., 2021). Re-forecasts are therefore bias corrected to counter this error. Our correction adheres to recommendations in Boer et al. (2016), who recommend an overarching bias correction method, regardless of the initialization type of the forecast. The reasons behind these recommendations are discussed in-depth in e.g. Boer et al. (2016), Kharin et al. (2012) and Hossain et al. (2022).
As explained above, the bias is model dependent and lead time dependent. Because of the assumption that the bias is different at each lead time, subtracting a mean drift over all times would lead to over-compensation at some lead times, and residual drift at

others. For this reason, we create lead-time based climatologies for each model. These are then used to create anomalies relative to the baseline climate.

In particular, I can't tell how to interpret the increased skill from reforecasts, in light of the differences in bias correction when using reforecast vs. GCM data to force the model. The reforecast data are bias corrected using different lead-time-based climatologies over 1971--2000. Different lead times aren't considered for the GCM-driven forecasts, so the GCM data have a single bias correction step using CRU TS data from 1961—1990. Are differences in prediction skill (i.e., simulated vs. observed mass balance) related to different bias correction methods, or the fact that reforecasts start from an observed climate field? Either would be useful to know about, but the authors don't address whether the bias correction has an effect.

Thank you for this comment. It is correct that lead time dependent bias correction is a fundamental step used in decadal forecasting, and so is mean bias correction in future projections for impact modeling (e.g. Lange, 2019) . The aim of our study is to compare the standard methodologies commonly applied in the respective fields. The GCM Historical data are processed as they would typically be in glacier modelling (e.g. Zekollari et al., 2024, Rounce et al., 2023, and many more), and the re-forecasts are also processed according to the recommendations mentioned above. It must be added that the drift correction following lead time (the length of time between the issuance of a forecast and the occurrence of the phenomena that were predicted) cannot be applied to the GCM Historical experiment where we have only one simulation.

We amended the text in several ways:
- Better clarify our goals as outlined above
- Better explain the drift correction, and explain why it can't be applied to the traditionally used GCM simulations
- In the discussion, mention and discuss the fact that drift correction does influence the results by a great amount and partly explains the performance of re-forecasts over traditional GCM-run simulations. The drift correction only impacts average-based metrics but not correlation or interannual skill.

These changes should hopefully help convey the main message of our study: decadal forecasts and the techniques developed to bias correct them should be preferred for medium range (e.g. decadal) predictions of glacier change over using historical simulations.

(also – why correct reforecasts over 1971—2000 and GCM data using 1961—1990 means? No explanation is given)

This is because the re-forecast DCPP project starts from 1961, so for 1961-1970, not all lead times are available. The GCM data is bias corrected as per OGGM standard, which is 1961-1990. To check that this doesn't change the results much, we ran an additional simulation with bias correction over the same period for the WGMS glaciers. A two-tailed t-test (significance level 0.05) reveals there are no statistically significant differences between the results.

3) Background on reforecasts and sources of skill

I think more background on initialized reforecasts is needed, to help the reader understand (i) the product being used to force the MB model, and (ii) where prediction skill might be coming from (if at all). I completely agree that decade timescales are of applied/operational importance, and this is an area worthy of focus. However, I found it puzzling that there is essentially no discussion of internal climate variability which is the main reason that forecasts on multi-annual to decade timeframes are challenging. The initialized climate models used for decadal forecasts are not summarized to much degree, or differentiated from GCMs, except for the fact that they are "initialized", but the authors do not really explain what is meant by "initialized". Again, this is key context when the main result is the relative skill of reforecast vs. GCM-driven mass balance predictions. Some physical reasoning for why an initialized forecast introduces more skill would be important for making sense of the results.

We agree, and hope that the following text and references help in this regard:

Decadal forecasts lie between numerical weather predictions and climate projections. Similar to numerical weather forecasts, they are initialized with current observations of the atmosphere, ocean, land surface and biochemistry, similar to numerical weather forecasting (Doblas-Reyes et al., 2013). The successful application of these methods to longer timescales, e.g. decadal, is an active area of research in operational climate prediction, which is a rapidly evolving field (Meehl et al., 2014; Smith et al., 2013). In particular at decadal timescales, initialization is expected to further improve our ability to detect the impact of radiative forcing induced trends on the occurrence and frequency of the internal climate modes of variability, such as the El Niño Southern Oscillation (ENSO) (O'Kane et al., 2019). This will contribute to accuracy in forecast temperature and precipitation, not present at the decadal timescale in (uninitialised) GCM projections.
Over the last years, larger initiatives on decadal prediction development have been set up, such as WCRP Grand Challenge on Near Term Climate Prediction (Kushnir et al., 2019). One of these initiatives is the Decadal Climate Prediction Project (DCPP), a coordinated multi-model investigation into decadal climate prediction, predictability, and variability, contributing to CMIP6 (Boer et al., 2016). The Decadal re-forecast experiment makes use of this DCPP output.

 We will amend the text accordingly.

Overall: thank you! These are all good points. We hope the background information above provides better context.

4) Visual examples of reforecasts

Finally, I think a figure showing some examples of the mass balance reforecasts would be helpful for understanding the method and results. The figures are largely aggregated statistics. Picking a glacier as a case study, and showing timeseries (perhaps individual members and ensemble means) of reforecasts under GCM vs. initialized forcing would help the reader immensely in understanding what the errors stats reported later would actually look like in terms of a forecast. It is great to draw on the wealth of WGMS data for validation, but I found myself wondering what these results actually look like for a given glacier. How quickly do the initialized reforecasts decorrelate due to internal variability? How do noise and trends compare? One or two examples would go a long way.

This is a very good idea and we have implemented it with an extra figure, also copied below. As can be seen, neither the GCM Historical nor the decadal re-forecast experiments perform very well when analyzed at the single glacier level, in this case the Hintereisferner and Langfjorjoekulen. In the Hintereisferner case, also cumulatively, neither the GCM Historical nor the Decadal re-forecasts have provided more skill than the very simple persistence experiment. The ensemble spread, in this case, is even larger than for the GCM Historical experiment, which does not speak for its benefits of initialization.

In the Langfjordjoekulen case, the decadal re-forecast experiment performs better than the GCM historical and persistence experiments. It does not follow the year-to-year variations, but captures the mean mass balance over the decade much better than the other two experiments.

[Figure]

[Figure]

*Fig. 3 Example for the Hintereisferner, Austria (above) and Langfjordjøkulen, Norway (below).*
*Mean errors (ME) for the different experiments, for mean mass balance over the decade, as in Table 2, are indicated in the legend. This means the difference between the mean observed mass balance and the mean simulated mass balance for the different experiments. For the GCM historical and decadal RF experiments, 'simulated' refers to the ensemble mean.*

Minor comments

114: What explains the pre-defined range of 50-600 for the melt factor? If outside of this is deemed not "physically realistic", what is assumed with an order-of-magnitude variation here? 50 kg m^-2 K^-1 seems very low – that's 0.05 m w.e. K^-1? At least a citation would be useful. Also note typo: K^-1 not K^1.

The typo has been corrected, thank you. The range mentioned above is what was used in all OGGM versions until 1.6. For these versions, OGGM did not apply a temperature correction to the reference historical data (e.g. CRU or ERA5) as is commonly done in some large scale models (e.g. Huss and Hock, 2015; Rounce et al., 2019). The melt factor therefore played the implicit role of temperature bias correction (Maussion et al., 2019, Sect. 3.3). "Physically realistic" here is therefore misleading and we will amend the text accordingly.

123: "which represent forecasting from very simple to complex". Some word is missing. Using simple to complex methods?

Yes, the words 'methods' was missing.

135: (and in general) If all of this is in terms of ensemble means, won't much of the absolute error in comparing observations to reforecasts come from the observation (a single timeseries) having more interannual variability than the ensemble mean? I think it can be valid to focus on ensemble means, but might need to alert the reader to this.

This is a good point, and will be discussed

136: For persistence forecasts with multi-year lead times, are the X years just repeated? Or is the mean used for forcing? I was unclear on how this works.

The X years are repeated. We will change the text accordingly.

166: Drift correction is mentioned here but hasn't been described yet, which may confuse a first-time reader.

True! We will refer to the section where it is explained

180: default correction factor – citation?

Citation added

235: errors in m w.e. – is that per year, or cumulative?

These are cumulative. Text will be changed.

244-45: "inability of a simple mass balance model to reliably simulate individual years". I don't think this has to do with the mass balance model… this is the inherent challenge of internal climate variability on these timescales.

Correct. See our response to the major comment above.

280-2: Isn't lower skill from single model ensembles here partly because the ensemble is smaller? Or are you comparing N=3 ensembles of either one of each, or 3 of the same? Please clarify.

Will be clarified in the manuscript. It indeed has to do with ensemble size

 Fig 4 caption: What is decadal forcing? decadal *reforecast* forcing?

Yes, decadal re-forecast forcing

359: What is "mean cumulative mass balance"? Seems contradictory. Or "mean" as in net annual balance?

The word 'mean' should have been, and will be, removed

365: SSP's drifting apart over a few to 10 years strikes me as a tiny effect comparted to internal variability and other factors on these timescales. See e.g., Hawkins and Sutton (2009).
This is true, and will be removed from the discussion

370: Climate change increasing the amplitude of natural variability is rather contentious, I would be hesitant to assert it as "likely" here. And I don't think this is the conclusion of Nijsse et al., 2019 – they are looking at the magnitude of variability across different models with different equilibrium climate sensitivities - which is different than the variability increasing as warming progresses.

This is a very fair assessment, and we agree that this assertion is too strong. We have will remove it from the discussion and are editing the section in accordance with the comments above and those by reviewer 1.

Reference
Hawkins, E., & Sutton, R. (2011). The potential to narrow uncertainty in projections of regional precipitation change. *Climate dynamics*, *37*, 407-418.

References:

Boer, G. J., Smith, D. M., Cassou, C., Doblas-Reyes, F., Danabasoglu, G., Kirtman, B., ... & Eade, R. (2016). The decadal climate prediction project (DCPP) contribution to CMIP6. *Geoscientific Model Development*, *9*(10), 3751-3777.

Delgado-Torres, C., Donat, M. G., Gonzalez-Reviriego, N., Caron, L. P., Athanasiadis, P. J., Bretonnière, P. A., ... & Doblas-Reyes, F. J. (2022). Multi-model forecast quality assessment of CMIP6 decadal predictions. *Journal of Climate*, *35*(13), 4363-4382.

Doblas-Reyes, F. J., García-Serrano, J., Lienert, F., Biescas, A. P., & Rodrigues, L. R. (2013). Seasonal climate predictability and forecasting: status and prospects. *Wiley Interdisciplinary Reviews: Climate Change*, *4*(4), 245-268.
Hossain, M. M., Garg, N., Anwar, A. F., Prakash, M., & Bari, M. (2022). Drift in CMIP5 decadal precipitation at catchment level. *Stochastic Environmental Research and Risk Assessment*, *36*(9), 2597-2616.

Huss, M., & Hock, R. (2015). A new model for global glacier change and sea-level rise. Frontiers in Earth Science, 3(September), 1–22. https://doi.org/10.3389/feart.2015.00054

Grieger, J., Smith, D., & Boer, G. (2016). Recommendations of the Decadal Climate Prediction Project for bias correction of decadal hindcasts.

Kharin, V. V., Boer, G. J., Merryfield, W. J., Scinocca, J. F., & Lee, W. S. (2012). Statistical adjustment of decadal predictions in a changing climate. *Geophysical Research Letters*, *39*(19).

Kushnir, Y., Scaife, A. A., Arritt, R., Balsamo, G., Boer, G., Doblas-Reyes, F., ... & Wu, B. (2019). Towards operational predictions of the near-term climate. *Nature Climate Change*, *9*(2), 94-101.

Lange, S.: Trend-preserving bias adjustment and statistical downscaling with ISIMIP3BASD (v1.0) (2019). *Geoscientific Model Development, 12,* https://doi.org/10.5194/gmd-12-3055-2019

Manzanas, R. (2020). Assessment of model drifts in seasonal forecasting: Sensitivity to ensemble size and implications for bias correction. *Journal of Advances in Modeling Earth Systems*, *12*(3), e2019MS001751.

Meehl, G. A., Goddard, L., Boer, G., Burgman, R., Branstator, G., Cassou, C., ... & Yeager, S. (2014). Decadal climate prediction: an update from the trenches. *Bulletin of the American Meteorological Society*, *95*(2), 243-267.

Mishra, N., Prodhomme, C., & Guemas, V. (2019). Multi-model skill assessment of seasonal temperature and precipitation forecasts over Europe. *Climate Dynamics*, *52*(7), 4207-4225.

Murphy, A. H. (1993). What is a good forecast? An essay on the nature of goodness in weather forecasting. *Weather and forecasting*, *8*(2), 281-293.

O'Kane, T. J., Sandery, P. A., Monselesan, D. P., Sakov, P., Chamberlain, M. A., Matear, R. J., ... & Stevens, L. (2019). Coupled data assimilation and ensemble initialization with application to multiyear ENSO prediction. *Journal of Climate*, *32*(4), 997-1024.

Pasternack, A., Grieger, J., Rust, H. W., & Ulbrich, U. (2021). Recalibrating decadal climate predictions–what is an adequate model for the drift?. *Geoscientific Model Development*, *14*(7), 4335-4355.

Rounce, D. R., Khurana, T., Short, M. B., Hock, R., Shean, D. E., & Brinkerhoff, D. J. (2020). Quantifying parameter uncertainty in a large-scale glacier evolution model using Bayesian inference: application to High Mountain Asia. Journal of Glaciology, 66(256), 175−187. https://doi.org/10.1017/jog.2019.91

Smith, D. M., Scaife, A. A., Boer, G. J., Caian, M., Doblas-Reyes, F. J., Guemas, V., ... & Wyser, K. (2013). Real-time multi-model decadal climate predictions. *Climate dynamics*, *41*, 2875-2888.

---

## Referee Report (RR1)

**Review of van der Laan et al. (2025) [10.5194/egusphere-2024-387](https://doi.org/10.5194/egusphere-2024-387)**

**General comments**

This manuscript by van der Laan et al. presents an impressive and relatively comprehensive analysis of the potential for global decadal glacier mass balance forecasts using bias-corrected CMIP6 data in OGGM. The authors compare the performance of OGGM in a re-forecast (also known as hindcast) setting when run with three different climate forcings: (1) an ensemble of initialized decadal re-forecasts extracted from CMIP6 DCPP-A, (2) baseline persistence forecasts that use forcing from previous years as a function of lead time, and (3) historical GCM simulations from an ensemble of uninitialized free-running CMIP6 outputs that represents the current state-of-the-art. Experiments are carried out on both a set of 279 so-called reference glaciers using high-quality WGMS glaciological mass balance data for calibration and for all $\simeq 214 \times 10^3$ land-terminating glaciers using geodetic mass balance data for calibration. With a fixed calibration routine it is convincingly shown that the decadal re-forecasts (1) can match or even outperform both the persistence forecasts (2) and historical GCM simulations (3) for ensemble-mean predictions of multi-year glacier mass balance. Crucially, by demonstrating the feasibility of global decadal glacier re-forecasts, this study paves the way for future work on global decadal glacier mass balance prediction that is vital for better capturing the near-term response of glaciers to ongoing anthropogenic global warming. This manuscript is both well written and structured with a clear experimental design. The results should be of great interest to the cryospheric science community, both for global glacier modeling and beyond. Moreover, it has already undergone an initial round of peer review with generally positive reviews. Although I was not involved in the earlier review process, overall I find myself in agreement with these mostly positive earlier comments. I recommend the publication of this manuscript once the minor and mostly technical points raised in the specific comments below have been addressed.

**Specific comments**

1. L2: Consider changing *"seasonal and long-term simulations"* to *"seasonal forecasts and long-term projections"* to be more precise. Simulations do not have to involve just forecasting/prediction, they could also be reanalyses that leverage historical observational data or scenario-based projections. Here, however, it becomes clear later that your focus is on demonstrating the potential of doing future decadal predictions through a re-forecasting exercise in the past two decades. I think this slight change in language would help situate your study and make your objectives clearer to the reader from the start of the abstract.

2. L37: Consider changing *"developmental"* to *"embryonic"* if you are trying to say that the field of decadal prediction (certainly for glaciers) is still early in its development.

3. L76: Consider changing *"the dynamical evolution of glaciers"* to just *"glacier flow"* since the term glacier dynamics is arguably vague even if it is commonly used in the field.

4. Correct Eq. 1 to
$$m_i(z) = p_f P_i^{\text{solid}} - \mu^* \text{max}\left(T_i(z) - T_{\text{melt}}, 0\right) + \epsilon \tag{1}$$

    In LaTeX code

    ```
    m_i(z)=p_f P_i^{\text{solid}}-\mu^*\mathrm{max}\left(T_i(z)-T_\text{melt},0\right)+\epsilon
    ```

    Corrections: (1) Text in the superscript of $P_i^{\text{solid}}$ and the subscript $T_{\text{melt}}$ should be in text (e.g. `mathrm{}`) mode. These should also be corrected in the text (L96, L97) (2) The max operator should also be in text mode and include the crucial second argument 0 to make it clear that it is the ramp function.

5. L96 z should be in math mode as $z$ (i.e. `$z$`).

6. L97: Avoid starting the sentence with a symbol change to (e.g.) *"Here the precipitation correction factor, $p_f$, is set to…"*.

7. Figure 2: This is a minor point and to some extent a matter of taste, but I would recommend transposing the axes here so that the observations (reference truth) are on the $x$-axis while the re-forecasts are on the $y$-axis. Such a change would ensure that *both* the sign and magnitude of the forecast−observation error correspond to the vertical direction and distance from the 1 : 1 line. That is, a positive error (overestimation) would coincide with a scatter point above the 1 : 1 line and vice-versa. To my knowledge, it also follows a (somewhat unwritten) convention for scatter plots when evaluating the performance of environmental models (Bennett et al., 2013; Pauwels and oters, 2019). Moreover, I would recommend going beyond the great first step of having equal axis limits by also making the axis aspect ratio equal so that the ticks on the $x$-axis and $y$-axis are equidistant.

8. L119: While I understand the reasoning for using the multi-model ensemble mean as a point estimate relying on some kind of wisdom of the crowd to get rid of outliers, it is surprising that you would completely disregard the ensemble spread as a measure of forecast uncertainty. One could argue that you are 'throwing the baby out with the bathwater' since by trying to get rid of outliers you are also disregarding the valuable uncertainty quantification inherent in an ensemble. While it is true that the ensemble spread here might be under-dispersive (overconfident), by choosing only the ensemble mean you are making your results degenerate and overconfident by design. I am not asking you to redo any analyses or add ensemble spread statistics, but I think this point should at least be discussed later in the paper in the discussion or outlook. Otherwise it leaves the reader wondering why you made this choice. Note that you are not actually getting rid of uncertainty by focusing on the mean, you are just hiding it. Why not embrace uncertainty and use probabilistic scores (Hersbach, 2000)?

9. L130: Here too superscripts should be in text mode, i.e. 21$^{\text{st}}$ not 21$^{st}$.

10. L138: Consider changing *"Per component of our study"* to *"For each component of our study"*.

11. L139: While I appreciate the emphasis on this point regarding the focus on forcing products rather than model calibration, in reality model calibration and the choice of forcing product exist in a state of strong entanglement. In particular, the absolute and relative performance of the 3 forecasting methods (re-forecast, persistence, GCM historical) may change as a function of the calibration method used. This would hold both in the case that the same calibrated parameters are used with each forcing product (but a different calibration routine is used) or if a different calibration is carried out for each forcing product. I understand that the focus of this study is *just* on the effect of the forcing product, but I would just like to push back a little by emphasizing that the fact that there is not really a clean separation between these two. In particular, the 'optimal' calibration parameters (two of which depend directly on the forcing) are conditional on the forcing product used. Ideally one would perform a sensitivity analysis of the joint (combined) effects of calibration routine and forcing product choice. Here too I am emphatically not asking you to redo any analyses or similar but I would recommend at least touching on this issue in an outlook section.

12. L153: Maybe I am missing something, but here you seemingly contradict yourselves. On the one hand, for component 2 you say that you do not need to use the residual $\epsilon$ since the dataset allows calibration with individual glaciers. On the other hand, for component 1 you do use the residual for individual glaciers (glacier-by-glacier basis). So which way around is it? Moreover, I wonder why you would ever not include a residual term in a calibration exercise without making strong assumptions. The residual represents observation and/or model errors and these are in reality never identically 0 even if this is sometimes assumed. Again, I am not asking you to change anything in the analysis but instead to clarify your assumptions.

13. L159: The wording here is somewhat unclear, presumably by 'perfect results' you mean that you are able to match the geodetic mass balance observations exactly. I guess this is not surprising if you have matched the number of parameters to the number of observations and if your model is flexible enough, but I would ask you to clarify what you mean by perfect in this sense do you fit the data on average or each datapoint (or are these the same). It is also not clear to me if you are fitting to a single data point estimating the geodetic mass balance from 2000 to 2020 or several within this period which section Section 3.2 seems to indicate. Perhaps this is obvious to frequent users of this dataset, but all readers should not be assumed to have this background knowledge. A final pedantic comment on this sentence is to consider changing *"mean bias"* to just *"bias"* since the term bias in statistics always refer to a mean (expected) error so the use of 'mean' here is redundant unless you are taking the mean of a mean but that would also require more explanation.

14. L187: Although this is already implicitly quite clear since you mention the reference height, I would nonetheless recommend specifying that this is *"air temperature"* to be more precise at least here when you introduce the climate forcing.

15. L241: Fix the LaTeX notation here, I guess something went wrong and $T't$ is supposed to be $\overline{T'_t}$ or similar.

16. L242: Change t to $t$ (i.e. `$t$`) for consistency.

17. L257: Be more specific here and change *"ensemble"* to *"ensemble mean"* since you do not consider the entire ensemble as a whole (only the mean) or probabilistic error metrics like the CRPS (Hersbach, 2000).

18. L260: In future work it would also be interesting to compare the ensemble skill (not just ensemble mean) of the re-forecasts to the historical GCM simulations. Perhaps something to allude to in an outlook. I suspect that the decadal re-forecasts have better calibrated uncertainty than the historical GCM simulations and this is something that could be quantified with probabilistic skill scores.

19. 272: Again, 'mean error' is synonymous with just 'bias' the term 'mean bias' is generally nonsensical. This term does sometimes appear in the literature since some researchers treat error and bias as synonymous, but the latter is strictly a statistic of the former.

20. L290: Consider clarifying what you are testing here. Is it the difference in the decadal mass balance estimates or the difference in skill (as measured by some metric) between the different experiments? I guess it is the latter, but if so, which skill metric are you comparing in the significance test?

21. L294: The procedure that you perform for the binomial test is also unclear. Did you do a binomial test for each glacier or a test based on statistics (e.g. the fraction with improved skill) across all 279 glaciers? More generally, consider dropping the overly dichotomous NHST framework involving thresholding in future studies and instead report the $p$-values (or statistics thereof) following recent recommendations McShane et al. (2019). More generally, be cautious of how to interpret the $p$-value and whether or not it is answering a statistical question that you are actually interested in (Ambaum, 2010).

22. L324: Change *"Analyzing"* to *"Comparing"*.

23. L330: This is a nice example of where reporting the $p$-values would make more sense rather than using the arbitrary traditional $p < 0.05$ threshold. In particular, the difference in significance here (or rather in the $p$-values) may itself not be significant (McShane et al., 2019). Maybe the $p$-values testing the difference in the decadal re-forecast and GCM historical experiment is not that different from the $p$-values testing the difference between the persistence and the other two experiments. The reader has no idea beyond the fact that the former are $p > 0.05$ (0.07 or 0.5?) while the latter is $p < 0.05$. Again, not something you have to change here, but in future work consider reporting $p$ on a continuous scale rather than an arbitrary threshold introduced haphazardly by a defunct statistician.

24. L380: Regional variations in the skill of simulated precipitation help explain the results, but also raise questions about the choice of a (uncalibrated) constant precipitation factor $p_f = 2.5$. I understand that (1) you are not focusing on calibration and (2) you hope to partly address the lack of skill in the climate simulations of precipitation by using a bias correction based on CRU. However, these CRU data are also coarse and will not necessarily add much skill to precipitation simulations, especially in complex topography. Although this is somewhat speculative, a more explicit joint downscaling/bias-correction to the glacier scale through calibration of $p_f$ could help add further skill to the re-forecasts. Several studies on seasonal snow (e.g. Fang et al., 2023) have shown that precipitation is the dominant source of uncertainty for seasonal snow storage in mountainous regions in global climate (reanalysis) products, and I would expect that this carries over at least partly to the glacier surface mass balance in some regions. I am not expecting you to redo any analyses at this advanced stage that are arguably outside the scope of the paper (since you do not want to focus on the calibration aspect), but I would nonetheless recommend touching on this precipitation calibration issue in the discussion as a potentially valuable topic to investigate in future studies.

25. L393: Consider being specific here and change *"historical simulations"* to *"GCM historical simulations"*.

26. L407: As previously alluded to, you had the opportunity to quantify whether or not there is an improvement in uncertainty quantification (both precision and accuracy) by (e.g.) comparing the CRPS (or some other score) of the glacier mass balance forced by an ensemble of re-forecasts to that forced by an ensemble of GCM historical simulations. As before, I am not asking or recommending you to do this in the present study but I am just highlighting the potential value of such an exercise in future work.

27. L422: I am fully in agreement. This further emphasizes the importance of reporting the $p$-values themselves rather than a binary significant/non-significant result.

28. L437: I agree that this could be an important step for future studies. As mentioned previously, I would also add (1) making full use of the uncertainty-aware ensemble of simulations (rather than just a point estimate from the ensemble mean) and (2) investigating the combined effects of calibration (also of $p_f$) and forcing data choice on the forecasts.

I would like to congratulate van der Laan et al. on their pioneering large scale glacier mass balance re-forecasting study and this well written manuscript which was a pleasure to read.
Kind regards,
Kristoffer Aalstad

**References**

Ambaum, M.: Significance Tests in Climate Science, J. Climate, https://doi.org/10.1175/2010JCLI3746.1, 2010.

Bennett, N. D. et al.: Characterising performance of environmental models, Environmental Modelling & Software, 40, 1–20, https://doi.org/https://doi.org/10.1016/j.envsoft.2012.09.011, 2013.

Fang, Y. et al.: Spatiotemporal snow water storage uncertainty in the midlatitude American Cordillera, TC, https://doi.org/10.5194/tc-17-5175-2023, 2023.

Hersbach, H.: Decomposition of the Continuous Ranked Probability Score for Ensemble Prediction Systems, Weather Forecasting, https://doi.org/10.1175/1520-0434(2000)015<0559:DOTCRP>2.0.CO;2, 2000.

McShane, B. et al.: Abandon Statistical Significance, The American Statistician, https://doi.org/https://doi.org/10.1080/00031305.2018.1527253, 2019.

Pauwels, V. R. and oters: Evaluating model results in scatter plots: A critique, Ecological Modelling, https://doi.org/10.1016/j.ecolmodel.2019.108802, 2019.

---

## Author Response (AR2)

Author response to reviews of "Decadal re-forecasts of glacier climatic mass balance" by van der Laan et al.

Dear editor and reviewer 2,

We would like to thank you for the thorough review and detailed comments on this second round of reviews. We appreciate the need for clarification and hope to offer it in the revised version. We propose to make changes to the manuscript in accordance with our replies below, in blue.

Thank you again for your time, kind regards,

Larissa van der Laan, on behalf of the author team

I have read the revised manuscript and appreciate the author's efforts to address many of the comments raised by myself and the other reviewer. However, I do think further clarifications are needed, especially around the statistical metrics. The level of significance (statistical and in terms of general impact) is still difficult to assess, partly because some of the metrics reported are unclear, and partly because I see a mismatch between the differences in skill shown, and the conclusions that the authors make. I am not totally convinced that reforecasts are as promising as the authors argue in some places. I'll grant that there could be reasonable differences of opinion on how promising one views reforecasts in light of these statistical results. But given that many results are found to statistically insignificant or marginal, at the very least the metrics for skill and the statistical tests used to evaluate them need to be much more clearly described, so the reader can make an informed interpretation of the results.

I elaborate on these major points of clarification below, and have some minor comments as well.

We appreciate this summary and will address both the major and minor comments below.

Major comments

1.) You say that you do these t-tests for individual glaciers. The wording is somewhat ambiguous – for a given comparison, are you doing

(A) 279 different t-tests (1 for each glacier)?

Or (B) a t-test based on the whole set of individual glaciers?

Wherever described (line 290, 304, 416), the wording seems to suggest (A), but then what are the samples being compared for each glacier's decadal mean mass balance? The 21-member ensembles? (But then how is this done for the persistence forecast which is a single member?). (B) seems to make more sense to me as a way to test the significance of the difference between approaches, so maybe I am misinterpreting the wording. But if it is (B) then some discussion would be warranted as to why the differences are not statistically significant

with a t-test, but are with a binomial test. I get that different statistical tests address different things, but then some explanation is needed for what these differences mean. Either way this should be clarified.

You are right, this is confusingly worded. We indeed did B), a single t-test comparing the mean performance across all 279 glaciers (ensemble means for the re-forecasts and historical GCMs, and the single member for persistence). We acknowledge that our original wording was ambiguous and have clarified this explicitly: "The samples for these tests are the glacier-specific values of decadal mean and cumulative mass balance for each forcing type. For decadal re-forecasts and GCM Historical experiments, these values are ensemble means, whereas for the persistence forecast, it is a single-member simulation.", see line 289.

The differences between the tests are discussed below, in comment 2.

2.) When discussing the binomial test, which metric is used to assess improved skill? MAE, ME, or correlation? If multiple, how are they combined? This is especially important to clarify as in table 2, while MAE and correlation indicate improved skill for reforecasts over the GCM approach, the model error (ME) metric is smaller in magnitude for the GCM Historical category. So it would seem important to comment on these differences across skill metrics, and what this means for the binomial test of improved skill.

We agree this needs clarification. The binomial test was based explicitly on the MAE metric only, as it directly measures the magnitude of errors and is most appropriate for evaluating forecast improvement across glaciers. ME and correlation were reported for completeness but not used for the binomial test. The clarification "We also perform a binomial test, assessing improved skill by specifically evaluating reductions in mean absolute error for the decadal re-forecast relative to persistence or historical forcing. This shows that out of the 279 glaciers we analyze…" can be found in line 293.

The differences between the tests and their interpretation is clarified as follows, see line 298:

"We note that the t-test and binomial test can yield different interpretations because they test different aspects of statistical significance. The t-test compares the mean differences between forecast methods across the entire set of glaciers, assessing whether differences in the average performance are statistically distinguishable given the spread in the dataset. By contrast, the binomial test assesses significance based only on the count of glaciers showing improved performance (in terms of lower mean absolute error) when comparing one forecast method directly against another, irrespective of the magnitude of improvement.

Thus, while the t-test might indicate no statistically significant difference due to variability across glaciers and the relatively small magnitude of improvement on average, the binomial test can indicate statistical significance simply because more glaciers improve under one approach compared to another than would be expected by chance alone. Both tests provide complementary perspectives: the t-test assesses the robustness of the average improvement magnitude across glaciers, whereas the binomial test evaluates consistency in improvement across individual glaciers."

3.) More generally, I find some of the conclusion statements about using initialized projections somewhat misleading or at odds with the magnitude/significance of improvements being shown. For instance, the abstract concludes with:

"These findings highlight the operational feasibility and significant potential of decadal predictions in glacier modeling for hydrological applications, particularly in regions where near-term forecasts can inform water resource management and climate adaptation strategies."

That is a strong statement to make for difference that in most cases are statistically insignificant. I think the small magnitude of the improvements should at least be noted in the abstract for transparency, and maybe the statistical testing briefly summarized (beyond the binomial result).

And later, at line 398: "While this has not translated into marked improvement for mass balance prediction in the current study, decadal forecasts' narrower ensemble spreads, because of their constraining the initial conditions of key climate variables, may reduce future uncertainty in near-term predictions critical for glacier response modeling"

This statement seems self-contradictory – why should we expect they may reduce future uncertainty, if there are not marked improvements shown here? I can't tell what is backing up this statement.

We do stand by this point, as the narrower ensemble spreads of decadal forecasts vs. long-term climate projections are one of the main strengths in terms of uncertainty reduction. We have clarified that this 'narrower spread' refers to decadal forecasts vs. projections, see line 409: "While we do not observe substantial improvement in mass balance predictions in this study, initialized decadal forecasts inherently provide narrower ensemble spreads due to their constrained initial conditions. Future improvements in initialization techniques, e.g. regarding the initialization of the North Atlantic Oscillation (Nicoli et al., 2025), may therefore still offer potential for reducing uncertainties in near-term glacier modeling, even if the current benefits are limited."

Overall, I think the authors should consider whether these conclusions are really warranted given the significance of results shown, and whether some of these statements ought to be adjusted.

We acknowledge the reviewer's overall point. Given the limited magnitude of the improvement and statistical significance, we've adjusted the abstract and conclusions to more clearly reflect the modest nature of the improvements, highlighting clearly the small absolute magnitudes and statistical significance.

The abstract conclusion has also been adjusted to: "These findings demonstrate moderate improvements from using decadal re-forecasts, though statistical significance is limited. While improvements are modest, these results suggest decadal re-forecasts may offer potential for improved near-term glacier predictions relevant for hydrological applications, particularly in regions where near-term forecasts can inform water resource management and climate adaptation strategies."
* * *
4.) I am not convinced by the argument at the end of section 3 that the improvements in skill may be greater than suggested by the t-tests. I do see the point – that the improvement is diluted by the shared skill coming from the global warming trend in both GCM and reforecast forcings. But that doesn't mean the actual skill improvements are larger, just potentially less likely to be spurious. At least reword to clarify. But if you think this test is not properly indicating the statistical significance, is there a better solution? Can the amount of skill coming from the global warming trend at least be quantified? I am not sure what the reader is supposed to do with this argument – it seems speculative to introduce this issue without doing anything about it.

We agree the wording was overly speculative. We've clarified that the issue is that statistical tests might underestimate the practical relevance (not the size) of improvements, rather than actual skill being greater than measured. Clarification in the text, see line 435:

"While this does not imply larger actual improvements, it does indicate that even modest skill increases from initialization may be reliably attributed to improved forecast initialization rather than random chance."

Minor comments:

There is one comment from my earlier review that I'm not sure was addressed. In sec 2.1, it is noted that final outputs are ensemble means. Isn't much of the absolute error in comparing observations to reforecasts coming from the observation (a single timeseries) having more interannual variability than the ensemble mean? And for this reason, if the output for persistence experiments is a single time series, again with more variability than the ensemble means for GCM and re-forecast experiments, how are we to compare the absolute error metrics these cases?

(The first response to this comment simply said it would be discussed but I did not see where it was discussed. I may have missed it, but please check)

This was indeed insufficiently addressed in the previous revision. The observational time series indeed contains greater interannual variability than the smoothed ensemble mean, with averaging multiple realizations naturally reduces variability by filtering. We acknowledge that this limitation somewhat favors ensemble-mean forecasts when evaluated via MAE compared to single-member persistence forecasts. However, ensemble means are commonly used in practice precisely because they aim to provide the most reliable prediction by averaging out unpredictable internal variability. Thus, comparing ensemble means to observations remains scientifically meaningful and justified, provided readers are clearly informed about this limitation. We hope to do this by adding the following beginning to the results and discussion section, see line 243:

"This section evaluates the results of the three experiments - decadal re-forecast, persistence, and GCM historical - by comparing them against observed glacier mass balance data for individual years and decadal averages. Throughout our analysis, we compare observational time series which inherently contain interannual variability to ensemble means from multi-member forecasts. It is important to recognize that ensemble means naturally reduce variability by averaging across multiple realizations, thereby smoothing internal fluctuations that occur in any single realization or observed series. This reduction in variability means that some portion of the absolute errors (e.g., the mean absolute error) used in the evaluation arises from the difference in variability between a single observational realization and the ensemble mean. Despite this limitation, ensemble means represent the most commonly used forecast product in practical applications due to their stability and reliability."

56 – what is "large scale glaciology"? I have a sense of what you mean, but not sure that's widely understood.

Clarified as "large-scale (regional to global) glacier modeling."

162 – specify – initial state of the climate/atmosphere? (As opposed to glacier)

Specified as "initial state of the climate system (atmosphere and ocean)"

181 – does a decreasing ensemble size have effects on the error metrics?

Explicity stated now that "Decreasing ensemble size at longer lead times typically increases forecast uncertainty, potentially affecting skill metrics slightly."

270 and Figure 2 – I still don't fully get what is added by comparing the 11 different overlapping decades. If those are getting averaged together in table 2, I think you would get the same #s by just averaging over 2000-2020, you are not adding any information by carving into 11 overlapping segments. And I still find the clusters of points in Figure 2 ambiguous to interpret since the individual points share a lot of information as overlapping decades. If it is just to demonstrate the effects of sampling error that is fine and useful, but not sure Figure 2 is needed for that.

We still believe it is important to show the overlapping decades, to illustrate variability in skill arising purely from changes in start and end years. These results highlight the sensitivity of decadal skill metrics to specific evaluation periods.
But figure 2 could in principle be moved to supplementary materials.

295 – why is the skill improvement different for decadal mean vs. cumulative? It seems odd that some glaciers would have improved skill for one and not the other. But as noted above, I'm not even sure which skill metric is used for the binomial test.

When presenting tables 2 and 3, are the skill metrics averaged across all glaciers? And across different decadal segments? Would be helpful to clarify.

This is indeed important to clarify, as we have done in line 272 : "The mean and cumulative decadal mass balances are distinct from each other as the cumulative mass balances only include the simulated years for which observations exist."

The skill metrics are indeed averaged across all glaciers and decadal segments.

382 – what counds as "good precipitation skill"?

Good precipitation skill refers to statistically significant correlation or low MAE between forecasted and observed precipitation in validation studies, such as Delgado-Torres et al. (2022)."

396 – I don't think there is any "predicting internal variability" – yes, there is still from starting with the right sampling of internal variability and capturing any climatic memory of that initial state… but I don't think predicting is the right word. And what is meant by regions where there is an externally forced response? I find this sentence confusing.

We hope this is more clear: "initialized forecasts better represent internal climate variability due to accurate initial conditions." See line 409

413 – I don't understand this sentence. What is the probabilistic context? And what are pathways? Ensemble members, or scenarios?

Probabilistic context refers to uncertainty arising from different socioeconomic emission scenarios (pathways) used in uninitialized projections.

419 – "this common signal…" I think this sentence is too similar to the sentence in the Smith et al., 2019 reference (p. 3). Either paraphrase or make a direct quote.

You're right, and we have indeed used a direct quote now.

430 – what counts as skillful here?

Skillful here is a summarizing word of the results presented in the manuscript, including the discussions of the error metrics.

446 – "rather than continuation of interdecadal trends" – this seems different than the persistence forecast, which is just repeating the same decade. Can this be clarified?

Yes, this is now clarified: "rather than depending solely on the continuation or repetition of recent decadal climatic conditions (as in persistence forecasts)." (Line 466)

Review of van der Laan et al. (2025) 10.5194/egusphere-2024-387

General comments

This manuscript by van der Laan et al. presents an impressive and relatively comprehensive analysis of the potential for global decadal glacier mass balance forecasts using bias-corrected CMIP6 data in OGGM. The authors compare the performance of OGGM in a re-forecast (also known as hindcast) setting when run with three different climate forcings: (1) an ensemble of initialized decadal re-forecasts extracted from CMIP6 DCPP-A, (2) baseline persistence forecasts that use forcing from previous years as a function of lead time, and (3) historical GCM simulations from an ensemble of uninitialized free-running CMIP6 outputs that represents the current state-of-the-art. Experiments are carried out on both a set of 279 so-called reference glaciers using high-quality WGMS glaciological mass balance data for calibration and for all $\simeq 214 \times 10^3$ land-terminating glaciers using geodetic mass balance data for calibration. With a fixed calibration routine it is convincingly shown that the decadal re-forecasts (1) can match or even outperform both the persistence forecasts (2) and historical GCM simulations (3) for ensemble-mean predictions of multi-year glacier mass balance. Crucially, by demonstrating the feasibility of global decadal glacier re-forecasts, this study paves the way for future work on global decadal glacier mass balance prediction that is vital for better capturing the near-term response of glaciers to ongoing anthropogenic global warming. This manuscript is both well written and structured with a clear experimental design. The results should be of great interest to the cryospheric science community, both for global glacier modeling and beyond. Moreover, it has already undergone an initial round of peer review with generally positive reviews. Although I was not involved in the earlier review process, overall I find myself in agreement with these mostly positive earlier comments. I recommend the publication of this manuscript once the minor and mostly technical points raised in the specific comments below have been addressed.

Dear editor and reviewer 1,

We would like to sincerely thank you for your detailed, thoughtful review and your constructive comments. They have greatly contributed to improving the manuscript's clarity and rigor and given us ideas for future studies. We have carefully considered each suggestion and made adjustments in the manuscript as detailed in our responses below ( in blue).

Thank you again for your valuable feedback and insights.

Kind regards,

Larissa van der Laan, on behalf of the author team

Specific comments

L2: Consider changing *"seasonal and long-term simulations"* to *"seasonal forecasts and long-term projections"* to be more precise. Simulations do not have to involve just forecasting/prediction, they could also be reanalyses that leverage historical observational data or scenario-based projections. Here, however, it becomes clear later that your focus is on demonstrating the potential of doing future decadal predictions through a re-forecasting exercise in the past two decades. I think this slight change in language would help situate your study and make your objectives clearer to the reader from the start of the abstract.

Agreed and revised to "seasonal forecasts and long-term projections" to improve clarity

L37: Consider changing *"developmental"* to *"embryonic"* if you are trying to say that the field of decadal prediction (certainly for glaciers) is still early in its development.

We changed this to 'early' rather than 'developmental', to reflect the early stage of the field

L76: Consider changing *"the dynamical evolution of glaciers"* to just *"glacier flow"* since the term glacier dynamics is arguably vague even if it is commonly used in the field.

Agreed, done

Correct Eq. 1 to

$$m_i(z) = p_f P_i^{\text{solid}} - \mu_* \max(T_i(z) - T_{\text{melt}}, 0) + \epsilon \qquad (1)$$

In LᴬTEXcode `m_i(z)=p_f P_i^{\text{solid}}-\mu^*\mathrm{max}\left(T_i(z)-T_\text{melt},0\right)+\epsilon`

Corrections: (1) Text in the superscript of $P_i^{\text{solid}}$ and the subscript $T_{\text{melt}}$ should be in text (e.g.

mathrm{}) mode. These should also be corrected in the text (L96, L97) (2) The max operator should also be in text mode and include the crucial second argument 0 to make it clear that it is the ramp function.

Done, and we thank the reviewer for being so considerate to add these precise suggestions to their review.

L96 z should be in math mode as $z$ (i.e. $z$).

Done

L97: Avoid starting the sentence with a symbol change to (e.g.) *"Here the precipitation correction factor, $p_f$, is set to..."*.

Done

Figure 2: This is a minor point and to some extent a matter of taste, but I would recommend transposing the axes here so that the observations (reference truth) are on the *x*-axis while the re-forecasts are on the *y*-axis. Such a change would ensure that *both* the sign and magnitude of the forecast−observation error correspond to the vertical direction and distance from the 1 : 1 line. That is, a positive error (overestimation) would coincide with a scatter point above the 1 : 1 line and vice-versa. To my knowledge, it also follows a (somewhat unwritten) convention for scatter plots when evaluating the performance of environmental models (Bennett et al., 2013; Pauwels and oters, 2019). Moreover, I would recommend going beyond the great first step of having equal axis limits by also making the axis aspect ratio equal so that the ticks on the *x*-axis and *y*-axis are equidistant.

We appreciate this comment and will take this convention into account for future manuscripts. For this case, we have ultimately decided to leave the figure as-is, as we do not feel the adjustments fundamentally change the figure's functionality.

L119: While I understand the reasoning for using the multi-model ensemble mean as a point estimate relying on some kind of wisdom of the crowd to get rid of outliers, it is surprising that you would completely disregard the ensemble spread as a measure of forecast uncertainty. One could argue that you are 'throwing the baby out with the bathwater' since by trying to get rid of outliers you are also disregarding the valuable uncertainty quantification inherent in an ensemble. While it is true that the ensemble spread here might be under-dispersive (overconfident), by choosing only the ensemble mean you are making your results degenerate and overconfident by design. I am not asking you to redo any analyses or add ensemble spread statistics, but I think this point should at least be discussed later in the paper in the discussion or outlook. Otherwise it leaves the reader wondering why you made this choice. Note that you are not actually getting rid of uncertainty by focusing on the mean, you are just hiding it. Why not embrace uncertainty and use probabilistic scores (Hersbach, 2000)?

We acknowledge this point regarding uncertainty quantification inherent in ensembles. We have expanded the

discussion to explicitly acknowledge that focusing solely on ensemble means indeed disregards uncertainty. We also in part address this in our response to reviewer 2, and have added a clarifying paragraph to the results/discussion section, see line 247. The outlook also proposes future use of probabilistic scores, see line 454.

L130: Here too superscripts should be in text mode, i.e. 21st not 21$^{st}$.

Done

L138: Consider changing *"Per component of our study"* to *"For each component of our study"*.

Done

L139: While I appreciate the emphasis on this point regarding the focus on forcing products rather than model calibration, in reality model calibration and the choice of forcing product

exist in a state of strong entanglement. In particular, the absolute and relative performance of the 3 forecasting methods (re-forecast, persistence, GCM historical) may change as a function of the calibration method used. This would hold both in the case that the same calibrated parameters are used with each forcing product (but a different calibration routine is used) or if a different calibration is carried out for each forcing product. I understand that the focus of this study is *just* on the effect of the forcing product, but I would just like to push back a little by emphasizing that the fact that there is not really a clean separation between these two. In particular, the 'optimal' calibration parameters (two of which depend directly on the forcing) are conditional on the forcing product used. Ideally one would perform a sensitivity analysis of the joint (combined) effects of calibration routine and forcing product choice. Here too I am emphatically not asking you to redo any analyses or similar but I would recommend at least touching on this issue in an outlook section.

We fully agree that model calibration and forcing product selection are interdependent. We have added a discussion to explicitly acknowledge that the calibration parameters are inherently conditional on the forcing product used and clearly highlight the value of exploring their combined effects in future research. See lin 400. That being said, this effect will in part already be reduced by the fact that we downscale the climate products to the baseline climate (CRU).

L153: Maybe I am missing something, but here you seemingly contradict yourselves. On the one hand, for component 2 you say that you do not need to use the residual $\epsilon$ since the dataset allows calibration with individual glaciers. On the other hand, for component 1 you do use the residual for individual glaciers (glacierby-glacier basis). So which way around is it? Moreover, I wonder why you would ever not include a residual term in a calibration exercise without making strong assumptions. The residual represents observation and/or model errors and these are in reality never identically 0 even if this is sometimes assumed. Again, I am not asking you to change anything in the analysis but instead to clarify your assumptions.

We do include a residual term for individual glaciers in component 1, but not in component 2, where individual glacier data allow for calibration without residuals: "[…] since this dataset consists of data for each glacier, removing the need for parameter transfer to glaciers without observations." (line 141) We also include a citation to Marzeion et al. (2012), which goes into more detail on the use of the residual.

L159: The wording here is somewhat unclear, presumably by 'perfect results' you mean that you are able to match the geodetic mass balance observations exactly. I guess this is not surprising if you have matched the number of parameters to the number of observations and if your model is flexible enough, but I would ask you to clarify what you mean by perfect in this sense do you fit the data on average or each datapoint (or are these the same). It is also not clear to me if you are fitting to a single data point estimating the geodetic mass balance from 2000 to 2020 or several within this period which section Section 3.2 seems to indicate. Perhaps this is obvious to frequent users of this dataset, but all readers should not be assumed to have this background knowledge. A final pedantic comment on this sentence is

to consider changing *"mean bias"* to just *"bias"* since the term bias in statistics always refer to a mean (expected) error so the use of 'mean' here is redundant unless you are taking the mean of a mean but that would also require more explanation.

Clarified "perfect results" explicitly as exactly matching the geodetic mass balance observations used for calibration. "This means that when run with the baseline climate CRU, it provides 'perfect results': exactly matching observations over the calibration period (bias of zero)." (line 148)

L187: Although this is already implicitly quite clear since you mention the reference height, I would nonetheless recommend specifying that this is *"air temperature"* to be more precise at least here when you introduce the climate forcing.

Agreed, done

—

L241: Fix the L$^A$TEX notation here, I guess something went wrong and $Tt$ is supposed to be $T'_t$ or similar.

True, thank you. Changed

L242: Change t to $t$ (i.e. $t$) for consistency.

Done

L257: Be more specific here and change *"ensemble"* to *"ensemble mean"* since you do not consider the entire ensemble as a whole (only the mean) or probabilistic error metrics like the CRPS (Hersbach, 2000).

Done

L260: In future work it would also be interesting to compare the ensemble skill (not just ensemble mean) of the re-forecasts to the historical GCM simulations. Perhaps something to allude to in an outlook. I suspect that the decadal re-forecasts have better calibrated uncertainty than the historical GCM simulations and this is something that could be quantified with probabilistic skill scores.

This is a good idea

272: Again, 'mean error' is synonymous with just 'bias' the term 'mean bias' is generally nonsensical. This term does sometimes appear in the literature since some researchers treat error and bias as synonymous, but the latter is strictly a statistic of the former.

True, and explained in 272. It is only explicitly mentioned because it is used in the literature cited in our work.

L290: Consider clarifying what you are testing here. Is it the difference in the decadal mass balance estimates or the difference in skill (as measured by some metric) between the different experiments? I guess it is the latter, but if so, which skill metric are you comparing in the significance test?

Clarified, see response to reviewer 2

L294: The procedure that you perform for the binomial test is also unclear. Did you do a binomial test for each glacier or a test based on statistics (e.g. the fraction with improved skill) across all 279 glaciers? More generally, consider dropping the overly dichotomous NHST framework involving thresholding in future studies and instead report the $p$-values (or statistics thereof) following recent recommendations McShane et al. (2019). More generally, be cautious of how to interpret the $p$-value and whether or not it is answering a statistical question that you are actually interested in (Ambaum, 2010).

Clarified, see response to reviewer 2. Thank you for the literature suggestions! Will take note of this for future publications.

L324: Change *"Analyzing"* to *"Comparing"*.

Done

L330: This is a nice example of where reporting the $p$-values would make more sense rather than using the arbitrary traditional $p < 0.05$ threshold. In particular, the difference in significance here (or rather in the $p$-values) may itself not be significant (McShane et al., 2019). Maybe the $p$-values testing the difference in the decadal re-forecast and GCM historical experiment is not that different from the $p$-values testing the difference between the persistence and the other two experiments. The reader has no idea beyond the fact that the former are $p > 0.05$ (0.07 or 0.5?) while the latter is $p < 0.05$. Again, not something you have to change here, but in future work consider reporting $p$ on a continuous scale rather than an arbitrary threshold introduced haphazardly by a defunct statistician.

Thank you for the suggestions and explanation, we will definitely keep this in mind for future work!

L380: Regional variations in the skill of simulated precipitation help explain the results, but also raise questions about the choice of a (uncalibrated) constant precipitation factor $p_f = 2.5$. I understand that (1) you are not focusing on calibration and (2) you hope to partly address the lack of skill in the climate simulations of precipitation by using a bias correction based on CRU. However, these CRU data are also coarse and will not necessarily add much skill to precipitation simulations, especially in complex topography. Although this is somewhat speculative, a more explicit joint downscaling/bias-correction to the glacier scale through calibration of $p_f$ could help add further skill to the re-forecasts. Several studies on seasonal snow (e.g. Fang et al., 2023) have shown that precipitation is the dominant source of uncertainty for seasonal snow storage in mountainous regions in global climate (reanalysis) products, and I would expect that this carries over at least partly to the glacier surface mass balance in some regions. I am not expecting you to redo any analyses at this advanced stage that are arguably outside the scope of the paper (since you do not want to focus on the

calibration aspect), but I would nonetheless recommend touching on this precipitation calibration issue in the discussion as a potentially valuable topic to investigate in future studies.

We fully agree with you, and this would be very interesting to include in future work. We appreciate your thinking and sharing these thoughts and ideas with us. It is now mentioned in the discussion as a potential future topic of research in this area. (see line 400)

L393: Consider being specific here and change *"historical simulations"* to *"GCM historical simulations"*.

Done

L407: As previously alluded to, you had the opportunity to quantify whether or not there is an improvement in uncertainty quantification (both precision and accuracy) by (e.g.) comparing the CRPS (or some other score) of the glacier mass balance forced by an ensemble of re-forecasts to that forced by an ensemble of GCM historical simulations. As before, I am not asking or recommending you to do this in the present study but I am just highlighting the potential value of such an exercise in future work.

Agreed!

L422: I am fully in agreement. This further emphasizes the importance of reporting the *p*-values themselves rather than a binary significant/non-significant result.

L437: I agree that this could be an important step for future studies. As mentioned previously, I would also add (1) making full use of the uncertainty-aware ensemble of simulations (rather than just a point estimate from the ensemble mean) and (2) investigating the combined effects of calibration (also of $p_f$) and forcing data choice on the forecasts.

Thanks for these thoughtful points

I would like to congratulate van der Laan et al. on their pioneering large scale glacier mass balance re-forecasting study and this well written manuscript which was a pleasure to read. Kind regards, Kristoffer Aalstad

Thank you for this detailed, comprehensive and immensely constructive review.

References

Ambaum, M.: Significance Tests in Climate Science, J. Climate, https://doi.org/10.1175/2010JCLI3746.1, 2010.

Bennett, N. D. et al.: Characterising performance of environmental models, Environmental Modelling & Software, 40, 1–20, https://doi.org/https://doi.org/10.1016/j.envsoft.2012.09.011, 2013.

Fang, Y. et al.: Spatiotemporal snow water storage uncertainty in the midlatitude American Cordillera, TC, https://doi.org/10.5194/tc-17-5175-2023, 2023.

Hersbach, H.: Decomposition of the Continuous Ranked Probability Score for Ensemble Prediction Systems, Weather Forecasting, https://doi.org/10.1175/1520-0434(2000)015<0559:DOTCRP>2.0.CO;2, 2000.

McShane, B. et al.: Abandon Statistical Significance, The American Statistician, https://doi.org/https://doi.org/ 10.1080/00031305.2018.1527253, 2019.

Pauwels, V. R. and oters: Evaluating model results in scatter plots: A critique, Ecological Modelling, https://doi.org/

10.1016/j.ecolmodel.2019.108802, 2019.

---

## Author Response (AR3)

Dear Larissa,

Thank you for being so patient while waiting for my delayed response.

I have read your responses to the reviews and the updated manuscript. I am satisfied that you have addressed these issues and clarified the text where needed. As a result, I am very happy to recommend the manuscript for acceptance.

There are some matters that I believe would clarify and strengthen the manuscript, which are related to the presentation and organization, and not any scientific content.

Congratulations and best wishes,
Ian

Dear Ian,

Thank you very much for your response! We've edited the paper according to the responses (in blue) below.

I'm grateful for your patience and guidance throughout this process! With kind regards, on behalf of the author team,

Larissa

Paragraph 306- 327-> What is the result of the t-test and binomial test in this paragraph?

To not overcomplicate matters, we've limited the results to 'significant difference' or 'not significant difference' for the t-test and for how many glaciers the result is improved for the binomial test.

Line 307-> "between" to amongst as there are three, not two cases.

Done

Consider a new paragraph for binomial tests, starting at line 308.

Done

Line 318 -> "across glaciers". It seems from 301, this is the set of glaciers from each experiment. Also in the previous paragraph, some values are significant. Can this be explained? Or is it due to the "might"? Maybe this sentence can be slightly clarified. Also, see next comment.

Clarified: *"Thus, while in some cases, the t-test does not indicate a statistically significant difference, due to variability across glaciers and the relatively small magnitude of improvement on average, the binomial test can indicate statistical significance simply*

*because more glaciers improve under one approach compared to another than would be expected by chance alone."*

Paragraph 318-322 -> should this be moved before 301? It might help clarify the previous point.

We tried both moving and not moving, but for text flow and clarity, decided against moving this and hope the altered sentence from the comment above is enough clarification.

For tables 4 and 5 -> would it make sense to add a row with the average of all RGI regions? This could make comparison easier amongst the three tests, and I believe that this would help support statements such as the one at lines 421-422

This is a good idea! We have added it.

Point 3 R1 -> Make sure point is referenced in Line 409. Recommend citation of the figure.

Done

Line 401-> Can a reference (i.e. figure/table) be provided for this statement?

Done

Ln 424-5 -> Can a referencing figure or finding be mentioned that supports this statement about the narrower ensemble spreads?

Edited the statement to be more specific to uncertainty rather than ensemble spread, and added citation: *"Initialized decadal forecasts also provide a more constant uncertainty over time than uninitialized projects, whose uncertainty increases significantly with lead time (Strobach et al., 2017). Future improvements in initialization, e.g. regarding the initialization of the North Atlantic Oscillation (Nicoli et al., 2025), may therefore still offer potential for reducing uncertainties in near-term glacier modeling, even if the current benefits are limited."*

Line 436-> "Despite these advantages, decadal forecasts are far from perfect, and the continuation of projects such as the DCPP contribution to CMIP6 (Boer et al., 2016) is essential to ensure their operational use."

Can this sentence be modified to either examine what specifically about these "forecasts are not perfect" or change the sentence?

Changed to: *"Despite these advantages, decadal forecasts are still in development, and the continuation of projects such as the DCPP contribution to CMIP6 (Boer et al., 2016) is essential to ensure their operational use."*

Line 437 -> "Current work" -> this work or other external work? Please cite if needed.

This work, changed

Line 443 -> "Statistically, we observe no significant difference between the decadal re-forecast and GCM Historical experiment per individual glacier." -> Referencing a figure or table in the text would strengthen this statement.

Done

Line 450 -> Should student be capitalized?

Yes, this has now been corrected.

Line 450 "While this does not imply larger actual improvement" -> improvement in model skill? Please specify the improvement.

Edited to: *"This poses that even modest skill increases from initialization may be reliably attributed to improved forecast initialization rather than random chance."*

Line 466 "The results shown here are limited by multiple factors and we especially highlight the need for continuing this research with a larger ensemble, which could increase predictive skill (Smith et al., 2013), and a more detailed look into ensemble spread, explicitly quantifying and incorporating ensemble spread as a measure of uncertainty." -> I think a verb is missing in the last part of this sentence. Would it be possible to shorten?

Agreed this was convoluted, we have turned into two sentences: "The results shown here are limited by multiple factors and we especially highlight the need for continuing this research with a larger ensemble, which could increase predictive skill (Smith et al., 2013). We also propose a more detailed look into ensemble spread, explicitly quantifying and incorporating ensemble spread as a measure of uncertainty.